# ViewFool: Evaluating the Robustness of Visual Recognition to Adversarial Viewpoints

**Yinpeng Dong**[1,3]**, Shouwei Ruan**[2]**, Hang Su**[1,4,5]**, Caixin Kang**[2]**, Xingxing Wei**[2]**, Jun Zhu**[1,3,4,5*]

[1] Dept. of Comp. Sci. and Tech., Institute for AI, Tsinghua-Bosch Joint ML Center,
THBI Lab, BNRist Center, Tsinghua University, Beijing 100084, China
[2] Institute of Artificial Intelligence, Beihang University, Beijing 100191, China
[3] RealAI    [4] Peng Cheng Laboratory    [5] Pazhou Laboratory (Huangpu), Guangzhou, China
`{dongyinpeng,suhangss,dcszj}@tsinghua.edu.cn, {shouweiruan,caixinkang,xxwei}@buaa.edu.cn`

## Abstract

Recent studies have demonstrated that visual recognition models lack robustness to distribution shift. However, current work mainly considers model robustness to 2D image transformations, leaving *viewpoint changes* in the 3D world less explored. In general, viewpoint changes are prevalent in various real-world applications (e.g., autonomous driving), making it imperative to evaluate viewpoint robustness. In this paper, we propose a novel method called ViewFool to find adversarial viewpoints that mislead visual recognition models. By encoding real-world objects as neural radiance fields (NeRF), ViewFool characterizes a distribution of diverse adversarial viewpoints under an entropic regularizer, which helps to handle the fluctuations of the real camera pose and mitigate the reality gap between the real objects and their neural representations. Experiments validate that the common image classifiers are extremely vulnerable to the generated adversarial viewpoints, which also exhibit high cross-model transferability. Based on ViewFool, we introduce ImageNet-V, a new out-of-distribution dataset for benchmarking viewpoint robustness of image classifiers. Evaluation results on 40 classifiers with diverse architectures, objective functions, and data augmentations reveal a significant drop in model performance when tested on ImageNet-V, which provides a possibility to leverage ViewFool as an effective data augmentation strategy to improve viewpoint robustness.

## 1 Introduction

One of the fundamental challenges of deep learning is to generalize reliably to unseen or shifted test distributions [16, 23, 44, 58]. Typically, deep learning models can be easily deceived by adversarial examples [19, 53], which are crafted by applying small perturbations to natural examples. Although the adversarial perturbations are imperceptible, they are quite sophisticated and can hardly exist in the wild [3, 30]. Recent works have revealed the vulnerability of visual recognition models to some natural (image) transformations, including rotation and translation [13], geometric transformations [27], image corruptions [23], and others [17, 31, 61, 62]. Out-of-distribution (OOD) generalization is thus emerging as an essential research topic in closing the gap between human and machine vision [16, 18].

Despite the progress, most works concentrate on the robustness of visual recognition models to 2D image transformations [13, 16, 23], while less effort has been made to explore model robustness to 3D variations in the physical world, such as *viewpoint changes*. In numerous real-world applications (e.g., autonomous driving, surveillance), viewpoint changes are arguably more natural and prevalent, which can cause severe security and safety consequences if the models cannot deal with such changes.

---

*Corresponding author.

36th Conference on Neural Information Processing Systems (NeurIPS 2022).

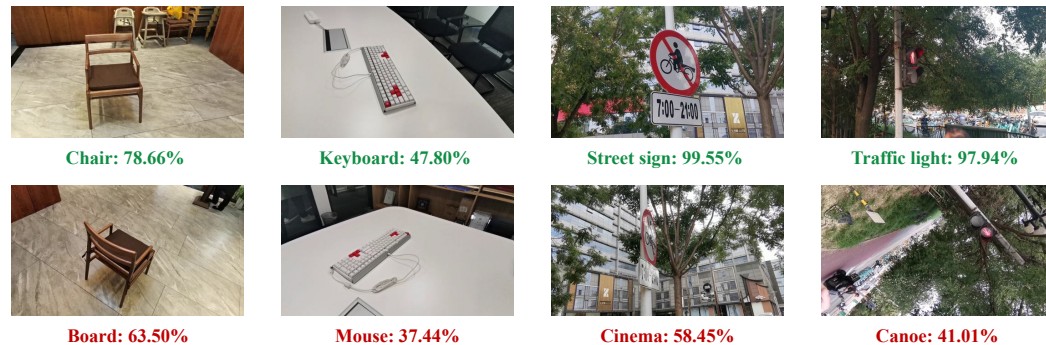

Figure 1: An illustration of adversarial viewpoints. **Top:** images captured from natural viewpoints, which are correctly classified; **Bottom:** the same objects taken from adversarial viewpoints found by ViewFool, which are wrongly classified. The target model is ResNet-50 [21].

As human vision exhibits a strong ability to robustly recognize objects from varying viewpoints [7], it is of significant importance to study the viewpoint robustness of visual recognition models to identify and understand their weaknesses before they are deployed in safety-critical applications.

Some prior works [5, 25] build new image datasets with different viewpoints to benchmark model robustness, but they cannot evaluate the *worst-case* performance of the model. Other works [2, 64] propose to adversarially estimate the physical conditions (e.g., viewpoints, illumination) that cause misclassification, but these methods can only perform attacks for synthetic 3D object models, which do not necessarily correspond to real-world objects. In general, evaluating the viewpoint robustness of visual recognition models in the physical world is considerably challenging, since it requires modeling the real-world 3D objects with high fidelity but there is always a gap between the real-world objects and their digital representations. Moreover, even if the adversarial viewpoints can be generated, it is hard to control the pose of the real camera to precisely match a specific adversarial viewpoint, making the captured image fail to consistently mislead the model in the physical world.

In this paper, we propose **ViewFool**, a novel method to systematically generate adversarial viewpoints that can mislead visual recognition models in the physical world, as shown in Fig. 1. Motivated by the recent progress of neural rendering, we represent the real-world 3D objects by neural radiance fields (NeRF) [41], which can synthesize photorealistic images from novel viewpoints. The rendered images are then fed into the target model to obtain the losses. By optimizing a parameterized distribution of adversarial viewpoints under an entropic regularizer, ViewFool can characterize diverse adversarial viewpoints to mitigate the gap between the real objects and their neural representations and improve the resistance to camera fluctuations. For optimization, we find that gradient calculation requires unacceptable GPU memory usage despite the differentiability of the rendering process (detailed in Sec. 3.3). Therefore, we resort to search gradients that only require forward propagation through the rendering process and the target model, making ViewFool applicable under a black-box setting with only query access to the model.

We conduct extensive experiments to evaluate the viewpoint robustness of image classifiers on the ImageNet [45] dataset. Our results demonstrate that ViewFool can effectively generate a distribution of adversarial viewpoints against the common image classifiers, which also exhibit high transferability across different models. Moreover, we introduce a new OOD benchmark—**ImageNet-V** for viewpoint robustness evaluation of image classifiers. We provide evaluation results on **40** image classifiers with diverse network architectures (e.g., ResNet [21], vision transformer [12]), training objective functions (e.g., self-supervised learning [22], adversarial training [46]), and data augmentation strategies (e.g., AugMix [24], DeepAugment [25]). The performance of these classifiers degrades significantly when evaluated on ImageNet-V, demonstrating that the existing classifiers are substantially vulnerable to viewpoint changes, which may motivate future work on improving viewpoint robustness.

## 2 Related work

**Deep learning robustness.** A major obstacle of deep learning models towards human-level performance is the lack of robustness to distribution shift [16, 18, 23, 44, 58]. It has been shown that deep

learning models are vulnerable to adversarial examples [19, 53], which are maliciously crafted by perturbing the natural examples with small perturbations. Some methods have successfully generated adversarial examples in the physical world [3, 30], raising a severe threat to the deployment of deep learning models. However, these adversarial examples are elaborately designed by humans, that can hardly exist in the wild. Other works have studied the vulnerability of deep learning models to natural transformations. For example, Engstrom et al. [13] find that image translations and rotations suffice to mislead a target model. Hendrycks & Dietterich [23] introduce ImageNet-C/P to benchmark model robustness to natural image corruptions and perturbations. The gap between human and machine vision for out-of-distribution generalization has also been analyzed [16, 18]. To address the robustness issues, various kinds of defense and robustness improvement techniques have been proposed, e.g., [24, 25, 33, 38, 63, 65], to name a few.

**Robustness to 3D variations.** As deep learning models have been increasingly deployed in numerous real-world applications, it is imperative to study model robustness in the 3D world. The ObjectNet [5] and DeepFashion Remixed [25] datasets have been introduced to evaluate model performance under varying backgrounds, camera viewpoints, and object rotations. However, they cannot evaluate the worst-case performance of the model to 3D variations. Zeng et al. [64] generate adversarial examples by perturbing the 3D physical conditions, including 3D rotations and translations, illuminations, etc. Alcorn et al. [2] further generate adversarial poses of the 3D objects, such that the rendered images can mislead deep learning models. However, these methods require synthetic 3D object models that do not correspond to real-world objects, and thus they cannot be utilized in the physical world. Our work differs from them mainly in that we can generate adversarial viewpoints for real-world objects by encoding them as neural representations.

**Novel view synthesis and neural rendering.** It is a long-standing problem to synthesize novel views of an object/scene given a set of captured images. Early methods [20, 32] can construct photorealistic novel views given densely sampled input views. Recently, plenty of works [14, 36, 39, 40, 41, 43, 50, 51] adopt neural networks to model 3D objects and can render high-quality images for sparser sets of input views. Among these approaches, neural radiance fields (NeRF) [39] show promising results by representing 3D objects/scenes as volumetric radiance fields parameterized by a multi-layer perceptron (MLP). In this paper, we adopt NeRF to learn neural representations of real-world 3D objects and then generate adversarial viewpoints based on the rendered images. Although NeRF can synthesize photorealistic images, there is still a reality gap between the rendered images and the real images, affecting the effectiveness of the adversarial viewpoints given the real objects. Our proposed method eliminates this issue by learning a distribution of adversarial viewpoints, as detailed below.

## 3 Methodology

In this section, we detail the proposed **ViewFool** method. ViewFool adopts NeRF to model real-world 3D objects as the digital representation and performs optimization in the search space of viewpoint parameters. In the following, we first introduce the background knowledge of NeRF and then present the problem formulation and optimization algorithm of ViewFool.

### 3.1 Preliminary: neural radiance fields (NeRF)

NeRF [41] encodes a real-world object/scene as a continuous volumetric radiance field $F : (\mathbf{x}, \mathbf{d}) \rightarrow (\mathbf{c}, \tau)$. $F$ is approximated by a multi-layer perceptron (MLP) which takes a 3D location $\mathbf{x} \in \mathbb{R}^3$ and a unit-norm viewing direction $\mathbf{d} \in \mathbb{R}^3$ as inputs, and outputs an emitted RGB color $\mathbf{c} \in [0, 1]^3$ and a volume density $\tau \in \mathbb{R}^+$. The volumetric radiance field can then be rendered into a 2D image from a specific viewpoint by computing the color of every pixel using volume rendering. Let $\mathbf{r}(t) = \mathbf{o} + t\mathbf{d}$ denote the camera ray emitted from the camera center $\mathbf{o}$ through a given pixel on the image plane. The color of this pixel can be approximated by

$$\hat{C}(\mathbf{r}) = \sum_{i=1}^{N} T(t_i) \cdot \alpha(\tau(t_i) \cdot \delta_i) \cdot \mathbf{c}(t_i), \quad \text{where } T(t_i) = \exp\left(-\sum_{j=1}^{i-1} \tau(t_j) \cdot \delta_j\right), \quad (1)$$

where a set of quadrature points $\{t_i\}_{i=1}^{N}$ is randomly selected by stratified sampling, $\mathbf{c}(t_i)$ and $\tau(t_i)$ are the color and density at $\mathbf{r}(t_i)$, $\alpha(x) = 1 - \exp(-x)$, and $\delta_i = t_{i+1} - t_i$ is the distance between two adjacent points.

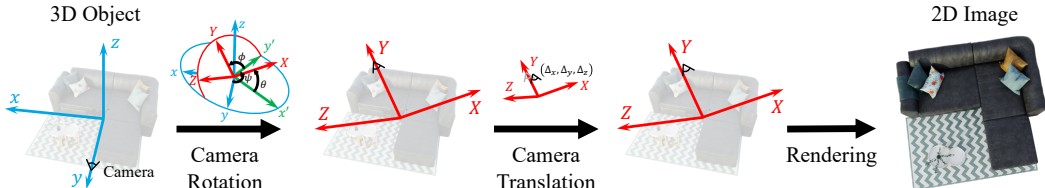

Figure 2: A demonstration of viewpoint changes. The camera is first rotated by $(\psi, \theta, \phi)$ angles about the $z$-$y'$-$X$ axes and then translated by $(\Delta_x, \Delta_y, \Delta_z)$. Given the new camera pose, a 2D image is rendered by NeRF and then fed to the target model for optimizing the viewpoint parameters.

To train the volumetric radiance field network $F$, NeRF minimizes the total squared error between the rendered and true pixel values given a set of images with known camera poses. NeRF also adopts the positional encoding and hierarchical volume sampling strategies to improve its performance. The network $F$ can leverage the multiview consistency among the calibrated images to implicitly capture the 3D nature of the underlying object, enabling it to render photorealistic novel views.

## 3.2 Problem formulation

Given a trained NeRF with the fixed parameters, ViewFool then estimates the adversarial viewpoints such that the rendered images can mislead a target visual recognition model $f$. As shown in Fig. 2, for a pre-defined coordinate system $xyz$ and the initialized camera pose, we first apply 3D rotations to the camera about the $z$-$y'$-$X$ axes in sequence by the Tait–Bryan angles $(\psi, \theta, \phi)$, which are also known as yaw, pitch, and roll. We then translate the camera position by $(\Delta_x, \Delta_y, \Delta_z)$ along the three axes. We let $\mathbf{v} := [\psi, \theta, \phi, \Delta_x, \Delta_y, \Delta_z] \in \mathbb{R}^6$ denote the transformation parameters of the camera. $\mathbf{v}$ is bounded in $[\mathbf{v}_{min}, \mathbf{v}_{max}]$ to make the captured images recognizable by humans. The setting of $\mathbf{v}_{min}$ and $\mathbf{v}_{max}$ will be specified in Sec. 4. Given the new camera pose, we can render a 2D image based on NeRF. Note that the rendered image is only determined by the viewpoint parameters $\mathbf{v}$, thus we denote it as $\mathcal{I} := \mathcal{R}(\mathbf{v})$, where $\mathcal{R}$ is the differentiable rendering process as introduced in Sec. 3.1.

Instead of searching for a single adversarial viewpoint, ViewFool captures a diverse set of adversarial viewpoints using a distribution $p(\mathbf{v})$. By optimizing the distribution $p(\mathbf{v})$ (under proper parameterization), any viewpoint $\mathbf{v}$ drawn from it is likely to fool the model. Therefore, we formulate ViewFool as solving the following optimization problem:

$$\max_{p(\mathbf{v})} \left\{ \mathbb{E}_{p(\mathbf{v})}[\mathcal{L}(f(\mathcal{R}(\mathbf{v})), y)] + \lambda \cdot \mathcal{H}(p(\mathbf{v})) \right\}, \tag{2}$$

where $f$ is an image classification model, $y$ is the ground-truth label of the object, $\mathcal{L}$ is the classification loss (e.g., cross-entropy loss), $\mathcal{H}(p(\mathbf{v})) = -\mathbb{E}_{p(\mathbf{v})}[\log p(\mathbf{v})]$ is the entropy of the distribution $p(\mathbf{v})$, and $\lambda$ is a balancing hyperparameter between the two loss terms. The entropic regularization is used to avoid the degeneration problem [11], i.e., $\max_{p(\mathbf{v})} \mathbb{E}_{p(\mathbf{v})}[\mathcal{L}(f(\mathcal{R}(\mathbf{v})), y)] \leq \max_{\mathbf{v}} \mathcal{L}(f(\mathcal{R}(\mathbf{v})), y)$, indicating that without the entropic regularization, the optimal distribution would degenerate into a Dirac one and cannot capture heterogeneous adversarial viewpoints. Therefore, ViewFool adopts the entropic regularization to address this issue with the following merits.

*First*, ViewFool can explore the space of adversarial viewpoints by learning the underlying distribution, which could help us better understand model vulnerabilities. *Second*, as it is hard to precisely control the real camera pose, learning a wider range of adversarial viewpoints can be more resistant to camera fluctuations, enabling to consistently mislead the target model in the physical world. *Third*, although NeRF shows promise for synthesizing photorealistic novel views of complex objects, there is still a gap between the real object and its neural representation, making the rendered images somewhat different from the real captured images. As a result, the adversarial viewpoint generated based on NeRF may not fool the model given the real object due to the appearance difference (which can be understood as a kind of adversarial noise introduced in NeRF rendering). ViewFool eliminates this issue by optimizing a continuous distribution of adversarial viewpoints because it is unlikely that the appearance difference can consistently fool the model under a variety of viewpoints if not optimized to do so [3]. *Fourth*, the entropic regularizer can help to avoid dropping into poor local optimal of the optimization problem to alleviate overfitting [10], leading to better cross-model transferability of the adversarial viewpoints. These benefits of ViewFool are validated by extensive experiments in Sec. 4.

## 3.3 Optimization algorithm

In ViewFool, we parameterize the distribution of adversarial viewpoints with trainable parameters. To define the distribution $p(\mathbf{v})$ whose support is contained in $[\mathbf{v}_{min}, \mathbf{v}_{max}]$, we take the transformation of random variable approach as

$$\mathbf{v} = \mathbf{a} \cdot \tanh(\mathbf{u}) + \mathbf{b}, \text{ where } \mathbf{u} \sim \mathcal{N}(\boldsymbol{\mu}, \boldsymbol{\sigma}^2 \mathbf{I}), \ \mathbf{a} = \frac{\mathbf{v}_{max} - \mathbf{v}_{min}}{2}, \ \mathbf{b} = \frac{\mathbf{v}_{max} + \mathbf{v}_{min}}{2}, \quad (3)$$

where $\mathbf{u}$ follows a diagonal Gaussian distribution with mean $\boldsymbol{\mu} \in \mathbb{R}^6$ and standard deviation $\boldsymbol{\sigma} \in \mathbb{R}^6$, and $\mathbf{v}$ is obtained by transforming $\mathbf{u}$ via $\tanh$ with proper normalization. Note that we utilize the diagonal Gaussian distribution in this paper for the sake of simplicity. Although the true distribution of adversarial viewpoints may not be Gaussian or unimodal, we could apply our method multiple times with different initializations or adopt mixture distributions. Our method is generally compatible with more expressive distributions, e.g., multiplicative normalizing flows [37] or diffusion probabilistic models [52], which we leave to future work. Given Eq. (3), problem (2) becomes

$$\max_{\boldsymbol{\mu}, \boldsymbol{\sigma}} \mathbb{E}_{\mathcal{N}(\mathbf{u}; \boldsymbol{\mu}, \boldsymbol{\sigma}^2 \mathbf{I})} \big[ \mathcal{L}(f(\mathcal{R}(\mathbf{a} \cdot \tanh(\mathbf{u}) + \mathbf{b})), y) - \lambda \cdot \log p(\mathbf{a} \cdot \tanh(\mathbf{u}) + \mathbf{b}) \big], \quad (4)$$

where the second term is the negative log density, whose expectation is the entropy $\mathcal{H}(p(\mathbf{v}))$.

To solve problem (4), we need to calculate the gradients of the loss w.r.t. the distribution parameters $(\boldsymbol{\mu}, \boldsymbol{\sigma})$. A common method to back-propagate the gradients from random samples to the distribution parameters is the low-variance reparameterization trick [8, 29]. In particular, we reparameterize $\mathbf{u}$ by $\mathbf{u} = \boldsymbol{\mu} + \boldsymbol{\sigma}\boldsymbol{\epsilon}$, where $\boldsymbol{\epsilon}$ is an auxiliary random variable following the standard Gaussian distribution $\mathcal{N}(\mathbf{0}, \mathbf{I})$. In principle, with the reparameterization, the gradients of the objective function in Eq. (4) w.r.t. $\boldsymbol{\mu}$ and $\boldsymbol{\sigma}$ can be calculated, since the NeRF rendering process $\mathcal{R}$ is naturally differentiable. However, we need to render the whole image of all pixels simultaneously, which requires a significant amount of GPU memory with a high image resolution. For example, it consumes more than 300G GPU memory for rendering a $400 \times 400$ image with $N = 128$ quadrature points each ray, making it impractical to directly calculate the gradients through the rendering process $\mathcal{R}$.

To address this problem, we adopt the search gradients to update the distribution parameters motivated by *natural evolution strategies* (NES) [60]. The gradients of the first loss in Eq. (4) can be derived as

$$\nabla_{\boldsymbol{\mu}, \boldsymbol{\sigma}} \mathbb{E}_{\mathcal{N}(\mathbf{u}; \boldsymbol{\mu}, \boldsymbol{\sigma}^2 \mathbf{I})} \big[ \mathcal{L}(f(\mathcal{R}(\mathbf{a} \cdot \tanh(\mathbf{u}) + \mathbf{b})), y) \big]$$
$$= \mathbb{E}_{\mathcal{N}(\mathbf{u}; \boldsymbol{\mu}, \boldsymbol{\sigma}^2 \mathbf{I})} \big[ \mathcal{L}(f(\mathcal{R}(\mathbf{a} \cdot \tanh(\mathbf{u}) + \mathbf{b})), y) \cdot \nabla_{\boldsymbol{\mu}, \boldsymbol{\sigma}} \log \mathcal{N}(\mathbf{u}; \boldsymbol{\mu}, \boldsymbol{\sigma}^2 \mathbf{I}) \big]. \quad (5)$$

Note that the gradients derived in Eq. (5) can be estimated with only forward propagation through the rendering process $\mathcal{R}$ and the classifier $f$, such that we do not need to render all pixels at the same time to reduce the memory usage. Moreover, it enables our method to operate in a *black-box* manner with only query access to the model. In practice, we adapt the plain search gradients by natural gradients to stabilize the optimization process [60].

For the second loss (i.e., entropy) in Eq. (4), we still adopt the reparameterization trick to analytically calculate its gradients to reduce the variance. Overall, the gradients of the objective function in Eq. (4) w.r.t. $\boldsymbol{\mu}$ and $\boldsymbol{\sigma}$ can be derived (in Appendix A) as

$$\nabla_{\boldsymbol{\mu}} = \mathbb{E}_{\mathcal{N}(\boldsymbol{\epsilon}; \mathbf{0}, \mathbf{I})} \left[ \mathcal{L}(f(\mathcal{R}(\mathbf{a} \cdot \tanh(\boldsymbol{\mu} + \boldsymbol{\sigma}\boldsymbol{\epsilon}) + \mathbf{b})), y) \cdot \boldsymbol{\sigma}\boldsymbol{\epsilon} - \lambda \cdot 2 \tanh(\boldsymbol{\mu} + \boldsymbol{\sigma}\boldsymbol{\epsilon}) \right], \quad (6)$$

$$\nabla_{\boldsymbol{\sigma}} = \mathbb{E}_{\mathcal{N}(\boldsymbol{\epsilon}; \mathbf{0}, \mathbf{I})} \left[ \mathcal{L}(f(\mathcal{R}(\mathbf{a} \cdot \tanh(\boldsymbol{\mu} + \boldsymbol{\sigma}\boldsymbol{\epsilon}) + \mathbf{b})), y) \cdot \frac{\boldsymbol{\sigma}(\boldsymbol{\epsilon}^2 - 1)}{2} - \lambda \cdot \frac{1 - 2\tanh(\boldsymbol{\mu} + \boldsymbol{\sigma}\boldsymbol{\epsilon}) \cdot \boldsymbol{\sigma}\boldsymbol{\epsilon}}{\boldsymbol{\sigma}} \right].$$

In practice, we approximate the expectation in gradient calculation by $k$ Monte Carlo (MC) samples. We perform multi-step gradient ascent to update the distribution parameters $(\boldsymbol{\mu}, \boldsymbol{\sigma})$ until convergence.

## 4 Experiments

We consider visual recognition models on ImageNet [45] in the experiments. To make the evaluation results fairer and more reproducible, we first conduct experiments on synthetic 3D object models in Sec. 4.1. We then study several aspects of ViewFool in Sec. 4.2. We present the experimental results of ViewFool on real-world 3D objects in Sec. 4.3. Finally, we introduce a new OOD dataset called **ImageNet-V** to benchmark viewpoint robustness. We provide evaluation results on **40** image classifiers with different architectures, training objective functions, and data augmentation strategies in Sec. 4.4. More results are provided in Appendix C. The code to reproduce the experimental results is publicly available at `https://github.com/Heathcliff-saku/ViewFool_`.

Table 1: The **attack success rates** of ViewFool on ResNet-50 and ViT-B/16. We consider viewpoint changes with camera translations only, camera rotations only, and the composition of camera translations and rotations. We present the attack success rates given the rendered images from the optimal distribution of adversarial viewpoints—$\mathcal{R}(p^*(\mathbf{v}))$, the rendered image from the optimal adversarial viewpoint—$\mathcal{R}(\mathbf{v}^*)$, and the real image from the optimal adversarial viewpoint—$\mathrm{Real}(\mathbf{v}^*)$.

| | Method | ResNet-50 | | | ViT-B/16 | | |
|---|---|---|---|---|---|---|---|
| | | $\mathcal{R}(p^*(\mathbf{v}))$ | $\mathcal{R}(\mathbf{v}^*)$ | $\mathrm{Real}(\mathbf{v}^*)$ | $\mathcal{R}(p^*(\mathbf{v}))$ | $\mathcal{R}(\mathbf{v}^*)$ | $\mathrm{Real}(\mathbf{v}^*)$ |
| Translation | Random Search | 38.08% | - | - | 26.01% | - | - |
| | ViewFool ($\lambda = 0$) | 54.27% | 60% | 44% | **46.30%** | 55% | 31% |
| | ViewFool ($\lambda = 0.01$) | **55.24%** | **67%** | **46%** | 43.10% | **59%** | **32%** |
| Rotation | Random Search | 51.68% | - | - | 45.60% | - | - |
| | ViewFool ($\lambda = 0$) | **89.62%** | 94% | 84% | **83.07%** | 87% | 78% |
| | ViewFool ($\lambda = 0.01$) | 84.25% | **96%** | **92%** | 79.25% | **91%** | **85%** |
| Translation + Rotation | Random Search | 53.98% | - | - | 47.52% | - | - |
| | ViewFool ($\lambda = 0$) | **92.01%** | 96% | 87% | **85.02%** | 91% | 77% |
| | ViewFool ($\lambda = 0.01$) | 88.79% | **98%** | **91%** | 82.14% | **92%** | **88%** |

## 4.1 Performance on synthetic 3D objects

**Experimental settings.** We collect 100 3D models of rigid objects from BlenderKit[2] belonging to the 1000 ImageNet classes to form our dataset. We do not adopt public datasets such as ShapeNet [9] due to the lack of realistic textures. More details about our dataset (including license, visualization) are provided in Appendix B. We provide the experiments on the more realistic Objectron dataset [1] in Appendix C.7. Following [41], we train a NeRF model for each 3D object using 100 images from varying viewpoints sampled on the upper hemisphere. We consider two image classification models, including the CNN-based ResNet-50 [21] and the Transformer-based ViT-B/16 [12]. They achieve 80.72% and 79.28% Top-1 accuracy on the naturally sampled images for training NeRF. In ViewFool, we initialize the camera at $[0, 4, 0]$ as shown in Fig. 2. We set the range of rotation angles as $\psi \in [-180°, 180°]$, $\theta \in [-30°, 30°]$, $\phi \in [20°, 160°]$, and the range of translation distances as $\Delta_x \in [-0.5, 0.5]$, $\Delta_y \in [-1, 1]$, $\Delta_z \in [-0.5, 0.5]$. We study viewpoint robustness of these models to camera translations only (with the fixed rotation angles $[\psi, \theta, \phi] = [0°, 0°, 65°]$), camera rotations only (with the fixed translation $[\Delta_x, \Delta_y, \Delta_z] = [0, 0, 0]$), and the composition of camera rotations and translations. We also study different ranges of rotation angles in Appendix C.5 and compare to adversarial 2D transformations in Appendix C.6. We set $\lambda = 0.01$ in the experiments and conduct an ablation study on $\lambda$ in Sec. 4.2. We approximate the gradients in Eq. (6) with $k = 50$ MC samples and adopt the Adam optimizer [28] to update the distribution parameters $(\boldsymbol{\mu}, \boldsymbol{\sigma})$ for 100 iterations.

**Evaluation metrics.** After obtaining the optimal parameters $(\boldsymbol{\mu}^*, \boldsymbol{\sigma}^*)$, we have the optimal distribution of adversarial viewpoints $p^*(\mathbf{v})$ as well as the mean of the distribution $\mathbf{v}^* = \mathbf{a} \cdot \tanh(\boldsymbol{\mu}^*) + \mathbf{b}$ with $\mathbf{a}$ and $\mathbf{b}$ defined in Eq. (3). $\mathbf{v}^*$ can be understood as the *optimal adversarial viewpoint*. We first measure the attack success rate (i.e., misclassification rate) of the classifier given the rendered image $\mathcal{R}(\mathbf{v}^*)$ by NeRF. We also calculate the attack success rate given the real image taken at the adversarial viewpoint $\mathbf{v}^*$ (denoted as $\mathrm{Real}(\mathbf{v}^*)$) for comparison. Moreover, to evaluate the performance of the learned distribution of adversarial viewpoints, we measure the attack success rate given the rendered images from 100 viewpoints sampled from $p^*(\mathbf{v})$, denoted as $\mathcal{R}(p^*(\mathbf{v}))$. All those attack success rates are averaged over all 100 objects in our dataset.

**Experimental results.** Table 1 shows the experimental results, where we compare ViewFool using $\lambda = 0.01$ with two baselines, including ViewFool without the entropic regularizer (i.e., $\lambda = 0$) and the random search method. For random search, we only report the results for $\mathcal{R}(p^*(\mathbf{v}))$ evaluated by 100 random viewpoints per object since there is no optimal viewpoint $\mathbf{v}^*$. We have the following findings based on the results.

**(1)** ViewFool with the entropic regularizer achieves higher attack success rates given the rendered image—$\mathcal{R}(\mathbf{v}^*)$ and the real image—$\mathrm{Real}(\mathbf{v}^*)$ from the optimal viewpoint $\mathbf{v}^*$ than that with $\lambda = 0$,

---

[2]https://www.blenderkit.com.

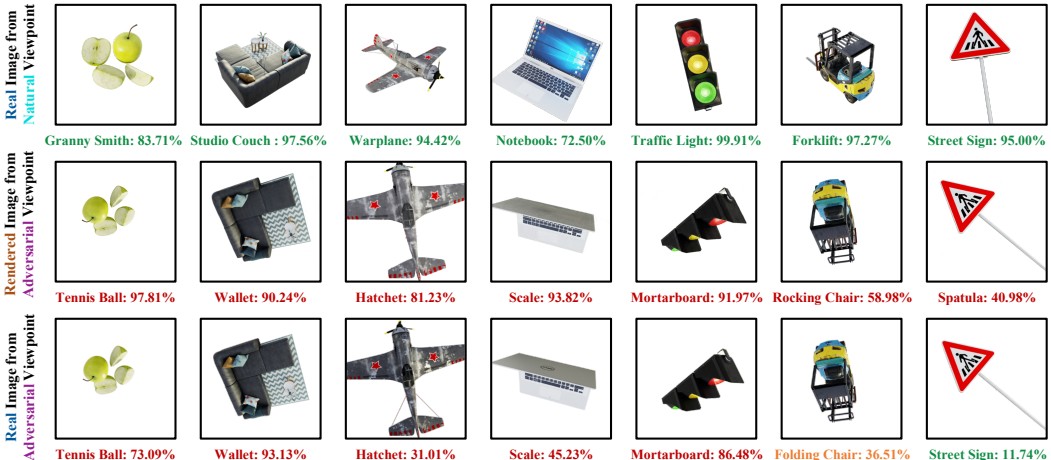

Figure 3: Visualization of the adversarial viewpoints generated by ViewFool against ResNet-50. The first row shows the real images taken from natural viewpoints that can be correctly classified. The second and third rows show the rendered images and the real images from adversarial viewpoints $\mathbf{v}^*$.

because the entropic regularizer enables to sufficiently explore the space of adversarial viewpoints and converge to better local optima.

**(2)** The performance gap between $\mathcal{R}(\mathbf{v}^*)$ and $\mathrm{Real}(\mathbf{v}^*)$ is smaller when using $\lambda = 0.01$ in most cases, indicating that ViewFool can better mitigate the reality gap between the rendered images and the real images. Thus, the generated adversarial viewpoints can consistently mislead the classifier.

**(3)** Using the entropic regularizer can result in lower attack success rates of $\mathcal{R}(p^*(\mathbf{v}))$, which is mainly because that the learned distributions with $\lambda = 0$ are more concentrated and cannot capture diverse adversarial viewpoints, as further verified in Table 2.

**(4)** We can see that optimizing the translation and rotation parameters simultaneously can lead to better performance than optimizing each of them alone. Thus, we consider the composition of camera translations and rotations in the following experiments.

**(5)** ViT-B/16 is more resistant to viewpoint changes than ResNet-50, showing the superiority of the transformer architecture, which is further validated with more architectures in Sec. 4.4.

**Visualization.** Fig. 3 shows the visualization results of the adversarial viewpoints $\mathbf{v}^*$ generated by ViewFool against ResNet-50. It can be seen that the real images taken from the natural viewpoints can be correctly classified by the model. By changing the viewpoints to $\mathbf{v}^*$, the rendered images and the real images are misclassified by the model to incorrect classes, while those images are still natural and recognizable for humans. In most cases, the model misclassifies the rendered image and the real image to the same class, although they appear differently in some details. However, there are some counter-examples that the rendered image and the real image are misclassified to different classes (see the second last column), or the real image is not misclassified (see the last column).

## 4.2 Additional results and ablation studies

**The effects of $\lambda$.** We adopt ResNet-50 as the target and consider $\lambda = 0, 0.0001, 0.001, 0.01, 0.1,$ and $1.0$, respectively. For each value of $\lambda$, we optimize the distribution of adversarial viewpoints by solving Eq. (4) for each object in our dataset. We measure the attack success rates given the rendered images from the optimal adversarial viewpoint $\mathcal{R}(\mathbf{v}^*)$ and the optimal distribution $\mathcal{R}(p^*(\mathbf{v}))$. To measure the diversity of the adversarial viewpoints, we further calculate the standard deviation of viewpoint parameters $[\psi, \theta, \phi, \Delta_x, \Delta_y, \Delta_z]$. The results are shown in Table 2. On one hand, we can observe that the standard deviation of every parameter increases along with $\lambda$, which is reasonable since we emphasize more on the entropic regularizer with a larger $\lambda$. On the other hand, as the distribution tends to cover a larger region of adversarial viewpoints, it inevitably contains unsuccessful viewpoints, leading to a reduction of attack success rate given $\mathcal{R}(p^*(\mathbf{v}))$. However, it does not affect

Table 2: The **attack success rates** and **standard deviation** of the each viewpoint parameter given $\lambda = 0, 0.0001, 0.001, 0.01, 0.1$, and $1.0$ in ViewFool. We choose ResNet-50 as the target model.

| | Attack Success Rate | | Standard Deviation | | | | | |
| | $\mathcal{R}(p^*(\mathbf{v}))$ | $\mathcal{R}(\mathbf{v}^*)$ | $\psi$ | $\theta$ | $\phi$ | $\Delta_x$ | $\Delta_y$ | $\Delta_z$ |
|---|---|---|---|---|---|---|---|---|
| $\lambda = 0$ | 92.01% | 96% | 4.554° | 1.681° | 2.474° | 0.031 | 0.064 | 0.029 |
| $\lambda = 0.0001$ | 90.69% | 97% | 6.797° | 2.130° | 3.385° | 0.040 | 0.077 | 0.037 |
| $\lambda = 0.001$ | 90.63% | 98% | 15.718° | 3.366° | 6.607° | 0.062 | 0.097 | 0.060 |
| $\lambda = 0.01$ | 88.79% | 98% | 21.644° | 3.397° | 7.485° | 0.065 | 0.099 | 0.060 |
| $\lambda = 0.1$ | 88.85% | 98% | 22.179° | 3.552° | 8.006° | 0.066 | 0.113 | 0.063 |
| $\lambda = 1.0$ | 77.00% | 92% | 23.937° | 4.193° | 9.209° | 0.071 | 0.141 | 0.069 |

Table 3: The **cross-model transferability** of the adversarial viewpoints against VGG-16, Inception-v3 (Inc-v3), Inception-ResNet-v2 (IncRes-v2), DenseNet-121 (DN-121), EfficientNet-B0 (EN-B0), MobileNet-v2 (MN-v2), DeiT-B, Swin-B, and Mixer-B. The adversarial viewpoints are sampled from the distribution learned by ViewFool with $\lambda = 0$ and $\lambda = 0.01$ against ResNet-50 and ViT-B/16.

| | ViewFool | VGG-16 | Inc-v3 | IncRes-v2 | DN-121 | EN-B0 | MN-v2 | DeiT-B | Swin-B | Mixer-B |
|---|---|---|---|---|---|---|---|---|---|---|
| ResNet-50 | $\lambda = 0$ | 85.00% | 75.94% | 80.59% | 73.97% | 76.73% | 76.77% | 65.22% | 55.81% | 87.07% |
| | $\lambda = 0.01$ | **86.52%** | **82.00%** | **82.00%** | **79.07%** | **82.62%** | **79.11%** | **69.35%** | **59.62%** | **90.37%** |
| ViT-B/16 | $\lambda = 0$ | 82.35% | 76.18% | 76.62% | 74.62% | **77.06%** | 72.14% | 69.34% | **60.50%** | **87.80%** |
| | $\lambda = 0.01$ | **82.83%** | **78.73%** | **79.07%** | **77.45%** | 74.92% | **73.97%** | **69.45%** | 59.01% | 85.72% |

the attack success rate given the rendered image $\mathcal{R}(\mathbf{v}^*)$ from the optimal viewpoint. As we discussed above, using an appropriate $\lambda > 0$ can achieve a higher attack success rate given $\mathcal{R}(\mathbf{v}^*)$.

**Cross-model transferability.** We then study the transferability of the generated adversarial viewpoints across different models. Specifically, we adopt 6 CNN-based models including VGG-16 [49], Inception-v3 [54], Inception-ResNet-v2 [55], DenseNet-121 [26], EfficientNet-B0 [56], MobileNet-v2 [47], 2 Transformer-based models including DeiT-B [59], Swin-B [35], and 1 MLP-based model MLP-Mixer [57]. We test the attack success rates of the adversarial viewpoints sampled from the optimal distribution $p^*(\mathbf{v})$ learned by ViewFool with $\lambda = 0$ and $\lambda = 0.01$ against ResNet-50 and ViT-B/16. We present the results in Table 3. The adversarial viewpoints have very high attack success rates against the black-box classifiers with different architectures, showing that the vulnerability of visual recognition models to viewpoint changes is prevalent and intrinsic among these models. Besides, using $\lambda = 0.01$ can achieve better cross-model transferability, since the entropic regularizer can avoid the overfitting problem [10] by escaping from poor local maxima of the optimization landscape.

**Resistance to camera fluctuations.** In the physical-world experiments, we cannot control the real camera pose to precisely match the adversarial viewpoints $\mathbf{v}^*$, thus it is important to ensure the resistance of the adversarial viewpoints to camera fluctuations. We simulate the physical-world experiments by changing the viewpoints within a percentage of camera fluctuations. Given a certain percentage $r\%$, we uniformly sample 20 random viewpoints from $[\mathbf{v}^* - (\mathbf{v}_{max} - \mathbf{v}_{min}) \cdot r\%, \mathbf{v}^* + (\mathbf{v}_{max} - \mathbf{v}_{min}) \cdot r\%]$ and test the attack success rates given the rendered images. Fig. 4 shows the curves of attack success rates of ViewFool against ResNet-50 with the percentage of fluctuations ranging from 1% to 10%. The results indicate that using $\lambda = 0.01$ can lead to better resistance to camera fluctuations that may be encountered in the physical world.

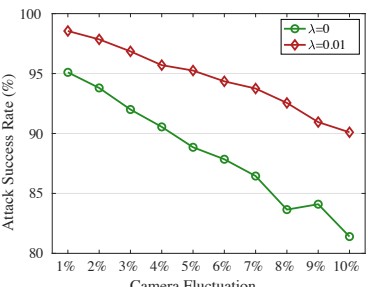

Figure 4: The curves of attack success rates of ViewFool given $\lambda = 0$ and $\lambda = 0.01$ w.r.t. fluctuations.

### 4.3 Performance on real-world 3D objects

In this section, we conduct two sets of real-world experiments to evaluate the performance of our method. The first set of experiments involves 8 real-world objects, including 2 warplanes, 1 pineapple,

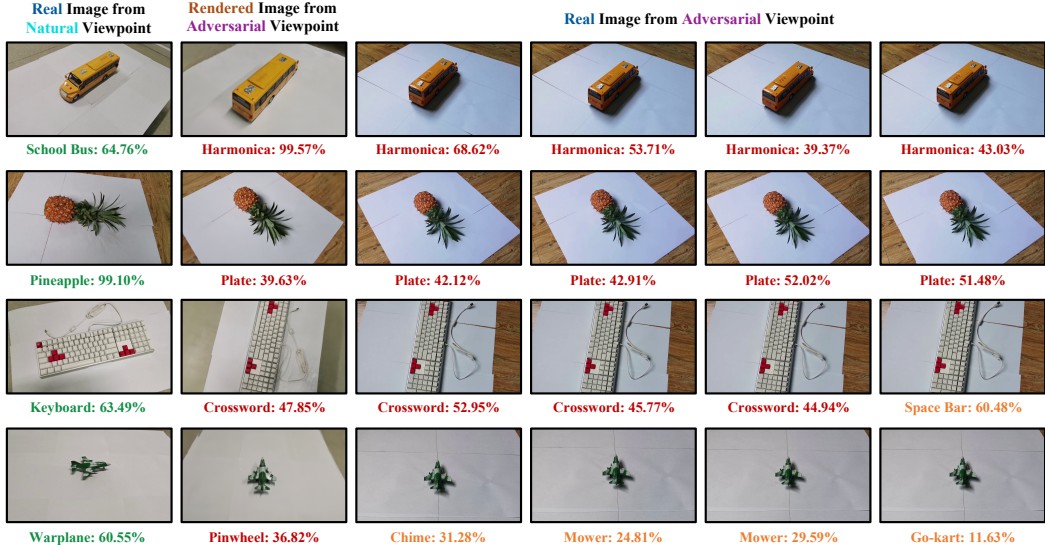

**Real** Image from **Natural** Viewpoint    **Rendered** Image from **Adversarial** Viewpoint      **Real** Image from **Adversarial** Viewpoint

| School Bus: 64.76% | Harmonica: 99.57% | Harmonica: 68.62% | Harmonica: 53.71% | Harmonica: 39.37% | Harmonica: 43.03% |
| Pineapple: 99.10% | Plate: 39.63% | Plate: 42.12% | Plate: 42.91% | Plate: 52.02% | Plate: 51.48% |
| Keyboard: 63.49% | Crossword: 47.85% | Crossword: 52.95% | Crossword: 45.77% | Crossword: 44.94% | Space Bar: 60.48% |
| Warplane: 60.55% | Pinwheel: 36.82% | Chime: 31.28% | Mower: 24.81% | Mower: 29.59% | Go-kart: 11.63% |

Figure 5: Visualization of real-world objects given different viewpoints. The first column shows the real images from natural viewpoints. The second column shows the rendered images from adversarial viewpoints. The 3-7 columns show the real images taken to approximate the adversarial viewpoints.

Table 4: The **attack success rates** of ViewFool in the physical world.

| Warplane #1 | Warplane #2 | Pineapple | School Bus | Keyboard | Shoe | Orange | French Loaf |
|---|---|---|---|---|---|---|---|
| 100% | 81% | 95% | 100% | 100% | 100% | 0% | 99% |

1 school bus, 1 keyboard, 1 shoe, 1 orange, and 1 French loaf. We train a NeRF model for each object given 100 captured images of resolution $1600 \times 900$ using a handheld cellphone. We place white paper underneath the object to simplify the background for NeRF training. We use COLMAP [48] to estimate the camera parameters of each image. After training NeRF, we adopt ViewFool with $\lambda = 0.01$ to optimize the distribution of adversarial viewpoints. The rendered images are resized and cropped to match the input size of the target model. We choose ResNet-50 and adopt the same hyperparameters as the previous experiments.

After obtaining the optimal adversarial viewpoint $\mathbf{v}^*$ for each object, we aim to take photos in the physical world to approximate $\mathbf{v}^*$, as shown in Fig. 5. Since we cannot precisely match the adversarial viewpoint $\mathbf{v}^*$, we take a video for each object around $\mathbf{v}^*$. We then extract 100 frames from the video to calculate the attack success rate in the physical world. The results are shown in Table 4. ViewFool can successfully mislead the target model on 7 out of the 8 objects, showing the effectiveness.

For the second set of experiments, we further consider another 4 objects, including two indoor objects (chair and keyboard) and two outdoor objects (street sign and traffic light). In this experiment, we do not place white paper underneath the object to be more realistic in the wild. We adopt the same experimental settings. Fig. 1 and Fig. C.4 in Appendix C.8 show the visualization results. ViewFool successfully generates adversarial viewpoints for all these 4 objects in the real world.

### 4.4 ImageNet-V benchmark

In this section, we introduce the ImageNet-V dataset for viewpoint robustness evaluation of image classifiers. Specifically, we render 100 images per object from varying viewpoints, which are sampled from the distribution $p^*(\mathbf{v})$ optimized by ViewFool with $\lambda = 0.01$. We choose ResNet-50 as the target model since it leads to better cross-model transferability than ViT-B/16, as shown in Table 3. Therefore, ImageNet-V consists of 10000 images of 100 different objects. Note that the background of all ImageNet-V images is white. This is conducive to evaluating viewpoint robustness solely.

We evaluate the performance of 40 image classifiers with different network architectures (including VGG [49], ResNet [21], Inception [54, 55], DenseNet [26], EfficientNet [56], MobileNet-v2 [47],

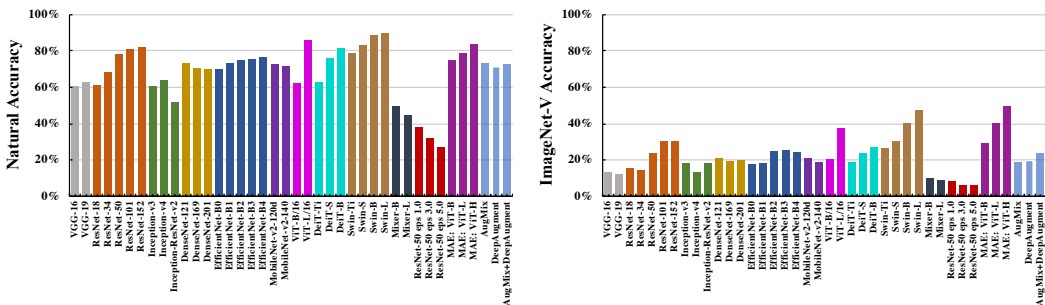

Figure 6: The classification accuracy of 40 image classifiers on images from natural viewpoints (left) and on ImageNet-V (right).

ViT [12], DeiT [59], Swin Transformer [35], and MLP Mixer [57]), objective functions (including adversarial training [46] and self-supervised MAE [22]), and data augmentation strategies (including AugMix [24] and DeepAugment [25]).

Fig. 6 shows the classification accuracy of these models on ImageNet-V. For comparison, we also evaluate their performance on 10000 images (100 per object) from natural viewpoints. Most classifiers achieve $70\% \sim 80\%$ accuracy on natural images, while none of them exceed $50\%$ accuracy on ImageNet-V, demonstrating that they are vulnerable to viewpoint changes. Among them, transformer-based models achieve the best performance (e.g., Swin-L obtains $47.40\%$ accuracy and MAE with ViT-H obtains $49.85\%$ accuracy), showing the superiority of the transformer architectures on OOD generalization [4, 6, 42]. Besides, a larger model size within the same architecture family tends to perform better. Adversarial training and data augmentation techniques, which show promise for adversarial and corruption robustness, do not obtain good results on ImageNet-V, demonstrating that ImageNet-V performance may not be correlated with that on other OOD robustness benchmarks.

## 5    Conclusion

This paper proposes ViewFool to generate adversarial viewpoints against image classification models. ViewFool adopts NeRF to represent real-world 3D objects and learns a distribution of adversarial viewpoints under an entropic regularizer for each object. Extensive experiments demonstrate the effectiveness of ViewFool and also reveal the vulnerability of common image classifiers to viewpoint changes. We successfully generated adversarial viewpoints for real-world 3D objects. Moreover, we established ImageNet-V for viewpoint robustness evaluation of any classifier. Besides evaluating viewpoint robustness, ViewFool has the possibility to be used as an effective data augmentation strategy to improve viewpoint robustness. A potential negative societal impact of ViewFool is that it could be used by malicious adversaries to cause security/safety issues in real-world applications.

**Limitations:** As an initial attempt for viewpoint robustness, this work also has several limitations. First, this work adopts a smaller dataset of 100 synthetic 3D objects for evaluation, which may be biased over all 1000 ImageNet classes, as elaborated in Appendix B. Second, our method requires high computational cost for NeRF training and optimizing adversarial viewpoints, as discussed in Appendix C.1. Third, this work only focuses on generating adversarial viewpoints (i.e., attacks) but does not propose a defense algorithm, which remains an open problem. We will address these limitations in the following work.

## Acknowledgement

This work was supported by the National Key Research and Development Program of China (No. 2017YFA0700904), NSFC Projects (Nos. 04132000422, 62061136001, 62076145, 62076147, U19B2034, U1811461, U19A2081, 61972224), Beijing NSF Project (No. JQ19016), BNRist (BNR2022RC01006), Tsinghua Institute for Guo Qiang, and the High Performance Computing Center, Tsinghua University. Y. Dong was also supported by the China National Postdoctoral Program for Innovative Talents and Shuimu Tsinghua Scholar Program. J. Zhu was also supported by the XPlorer Prize.

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
