# A  Proof of Eq. (6)

In this section, we derive the gradients of the objective in Eq. (4) w.r.t. the distribution parameters $(\boldsymbol{\mu}, \boldsymbol{\sigma})$.

For the first loss $\mathcal{L}_1 := \mathbb{E}_{\mathcal{N}(\mathbf{u};\boldsymbol{\mu},\boldsymbol{\sigma}^2\mathbf{I})}[\mathcal{L}(f(\mathcal{R}(\mathbf{a} \cdot \tanh(\mathbf{u}) + \mathbf{b})), y)]$ in Eq. (4), We adopt the search gradients as shown in Eq. (5), in which we can derive that

$$\nabla_{\boldsymbol{\mu}} \log \mathcal{N}(\mathbf{u}; \boldsymbol{\mu}, \boldsymbol{\sigma}^2\mathbf{I}) = \frac{\mathbf{u} - \boldsymbol{\mu}}{\boldsymbol{\sigma}^2} = \frac{\boldsymbol{\epsilon}}{\boldsymbol{\sigma}}; \tag{A.1}$$

$$\nabla_{\boldsymbol{\sigma}} \log \mathcal{N}(\mathbf{u}; \boldsymbol{\mu}, \boldsymbol{\sigma}^2\mathbf{I}) = \frac{(\mathbf{u} - \boldsymbol{\mu})^2 - \boldsymbol{\sigma}^2}{\boldsymbol{\sigma}^3} = \frac{\boldsymbol{\epsilon}^2 - 1}{\boldsymbol{\sigma}}, \tag{A.2}$$

where $\epsilon$ follows the standard Gaussian distribution.

As we mentioned in Sec. 3.3, we adapt the plain search gradients by natural gradients to stabilize the optimization process, as suggested in [60]. The natural gradient is defined as

$$\widetilde{\nabla}_{\boldsymbol{\mu},\boldsymbol{\sigma}} \mathcal{L}_1 = \mathbf{F}^{-1} \nabla_{\boldsymbol{\mu},\boldsymbol{\sigma}} \mathcal{L}_1, \tag{A.3}$$

where $\mathbf{F}$ is the Fisher information matrix as

$$\mathbf{F} = \mathbb{E}_{\mathcal{N}(\mathbf{u};\boldsymbol{\mu},\boldsymbol{\sigma}^2\mathbf{I})} \left[ \nabla_{\boldsymbol{\mu},\boldsymbol{\sigma}} \log \mathcal{N}(\mathbf{u}; \boldsymbol{\mu}, \boldsymbol{\sigma}^2\mathbf{I}) \cdot \nabla_{\boldsymbol{\mu},\boldsymbol{\sigma}} \log \mathcal{N}(\mathbf{u}; \boldsymbol{\mu}, \boldsymbol{\sigma}^2\mathbf{I})^\top \right]. \tag{A.4}$$

Therefore, we can derive that

$$\mathbf{F}_{\boldsymbol{\mu}} = \frac{\mathbf{I}}{\boldsymbol{\sigma}^2}, \quad \widetilde{\nabla}_{\boldsymbol{\mu}} \log \mathcal{N}(\mathbf{u}; \boldsymbol{\mu}, \boldsymbol{\sigma}^2\mathbf{I}) = \mathbf{F}_{\boldsymbol{\mu}}^{-1} \nabla_{\boldsymbol{\mu}} \log \mathcal{N}(\mathbf{u}; \boldsymbol{\mu}, \boldsymbol{\sigma}^2\mathbf{I}) = \boldsymbol{\sigma}\boldsymbol{\epsilon}; \tag{A.5}$$

$$\mathbf{F}_{\boldsymbol{\sigma}} = \frac{2 \cdot \mathbf{I}}{\boldsymbol{\sigma}^2}, \quad \widetilde{\nabla}_{\boldsymbol{\sigma}} \log \mathcal{N}(\mathbf{u}; \boldsymbol{\mu}, \boldsymbol{\sigma}^2\mathbf{I}) = \mathbf{F}_{\boldsymbol{\sigma}}^{-1} \nabla_{\boldsymbol{\sigma}} \log \mathcal{N}(\mathbf{u}; \boldsymbol{\mu}, \boldsymbol{\sigma}^2\mathbf{I}) = \frac{\boldsymbol{\sigma}(\boldsymbol{\epsilon}^2 - 1)}{2}. \tag{A.6}$$

By integrating Eq. (A.5) and Eq. (A.6) into Eq. (5), we can observe the gradients of the first loss as

$$\widetilde{\nabla}_{\boldsymbol{\mu}} \mathcal{L}_1 = \mathbb{E}_{\mathcal{N}(\boldsymbol{\epsilon};\mathbf{0},\mathbf{I})}[\mathcal{L}(f(\mathcal{R}(\mathbf{a} \cdot \tanh(\boldsymbol{\mu} + \boldsymbol{\sigma}\boldsymbol{\epsilon}) + \mathbf{b})), y) \cdot \boldsymbol{\sigma}\boldsymbol{\epsilon}]; \tag{A.7}$$

$$\widetilde{\nabla}_{\boldsymbol{\sigma}} \mathcal{L}_1 = \mathbb{E}_{\mathcal{N}(\boldsymbol{\epsilon};\mathbf{0},\mathbf{I})} \left[ \mathcal{L}(f(\mathcal{R}(\mathbf{a} \cdot \tanh(\boldsymbol{\mu} + \boldsymbol{\sigma}\boldsymbol{\epsilon}) + \mathbf{b})), y) \cdot \frac{\boldsymbol{\sigma}(\boldsymbol{\epsilon}^2 - 1)}{2} \right]. \tag{A.8}$$

For the second loss $\mathcal{H} := \mathbb{E}_{\mathcal{N}(\mathbf{u};\boldsymbol{\mu},\boldsymbol{\sigma}^2\mathbf{I})}[-\log p(\mathbf{a} \cdot \tanh(\mathbf{u}) + \mathbf{b})]$, we take the transformation of random variable approach to rewrite $\mathcal{H}$ as $\mathcal{H} = \mathbb{E}_{\mathcal{N}(\boldsymbol{\epsilon};\mathbf{0},\mathbf{I})}[-\log p(\mathbf{a} \cdot \tanh(\boldsymbol{\mu} + \boldsymbol{\sigma}\boldsymbol{\epsilon}) + \mathbf{b})]$. The log density of the distribution can be analytically calculated as follows.

Note that each dimension of the random variable is independent, thus we only consider one dimension. The density of $\epsilon$ is $p(\epsilon) = \frac{1}{\sqrt{2\pi}} \exp(-\frac{\epsilon^2}{2})$. The density of $u := \mu + \sigma\epsilon$ is $p(u) = \frac{1}{\sqrt{2\pi}\sigma} \exp(-\frac{\epsilon^2}{2})$. Let $v := a\tanh(u) + b$, then the inverse transformation is $u = \tanh^{-1}(\frac{v-b}{a}) = \frac{1}{2}\log(\frac{a+v-b}{a-v+b})$. The derivative of $u$ w.r.t. $v$ is $\frac{1}{a(1-\tanh(u)^2)}$. By applying the transformation of variable approach, we have the density of $v$ as

$$p(v) = \frac{1}{\sqrt{2\pi}\sigma} \exp(-\frac{\epsilon^2}{2}) \frac{1}{a(1 - \tanh(u)^2)} = \frac{1}{\sqrt{2\pi}\sigma} \exp(-\frac{\epsilon^2}{2}) \frac{1}{a(1 - \tanh(\mu + \sigma\epsilon)^2)}. \tag{A.9}$$

Hence, the negative log density of $p(v)$ is

$$-\log p(v) = \frac{\epsilon^2}{2} + \frac{\log(2\pi)}{2} + \log\sigma + \log(1 - \tanh(\mu + \sigma\epsilon)^2) + \log a. \tag{A.10}$$

Sum over all dimensions, we have

$$\mathcal{H} = \mathbb{E}_{\mathcal{N}(\boldsymbol{\epsilon};\mathbf{0},\mathbf{I})} \left[ \sum_d \left( \frac{\boldsymbol{\epsilon}_d^2}{2} + \frac{\log(2\pi)}{2} + \log\boldsymbol{\sigma}_d + \log(1 - \tanh(\boldsymbol{\mu}_d + \boldsymbol{\sigma}_d\boldsymbol{\epsilon}_d)^2) + \log\mathbf{a}_d \right) \right], \tag{A.11}$$

where $d$ is the dimension index. Given Eq. (A.11), we can simply calculate the gradients of $\mathcal{H}$ w.r.t. $\boldsymbol{\mu}$ and $\boldsymbol{\sigma}$ as

$$\nabla_{\boldsymbol{\mu}} \mathcal{H} = \mathbb{E}_{\mathcal{N}(\boldsymbol{\epsilon};\mathbf{0},\mathbf{I})}[-2\tanh(\boldsymbol{\mu} + \boldsymbol{\sigma}\boldsymbol{\epsilon})]; \tag{A.12}$$

$$\nabla_{\boldsymbol{\sigma}} \mathcal{H} = \mathbb{E}_{\mathcal{N}(\boldsymbol{\epsilon};\mathbf{0},\mathbf{I})} \left[ \frac{1 - 2\tanh(\boldsymbol{\mu} + \boldsymbol{\sigma}\boldsymbol{\epsilon}) \cdot \boldsymbol{\sigma}\boldsymbol{\epsilon}}{\boldsymbol{\sigma}} \right]. \tag{A.13}$$

As the overall loss is $\mathcal{L}_1 + \lambda \cdot \mathcal{H}$, by combining Eq. (A.7) and Eq. (A.12), Eq. (A.8) and Eq. (A.13), we obtain the gradients in Eq. (6).

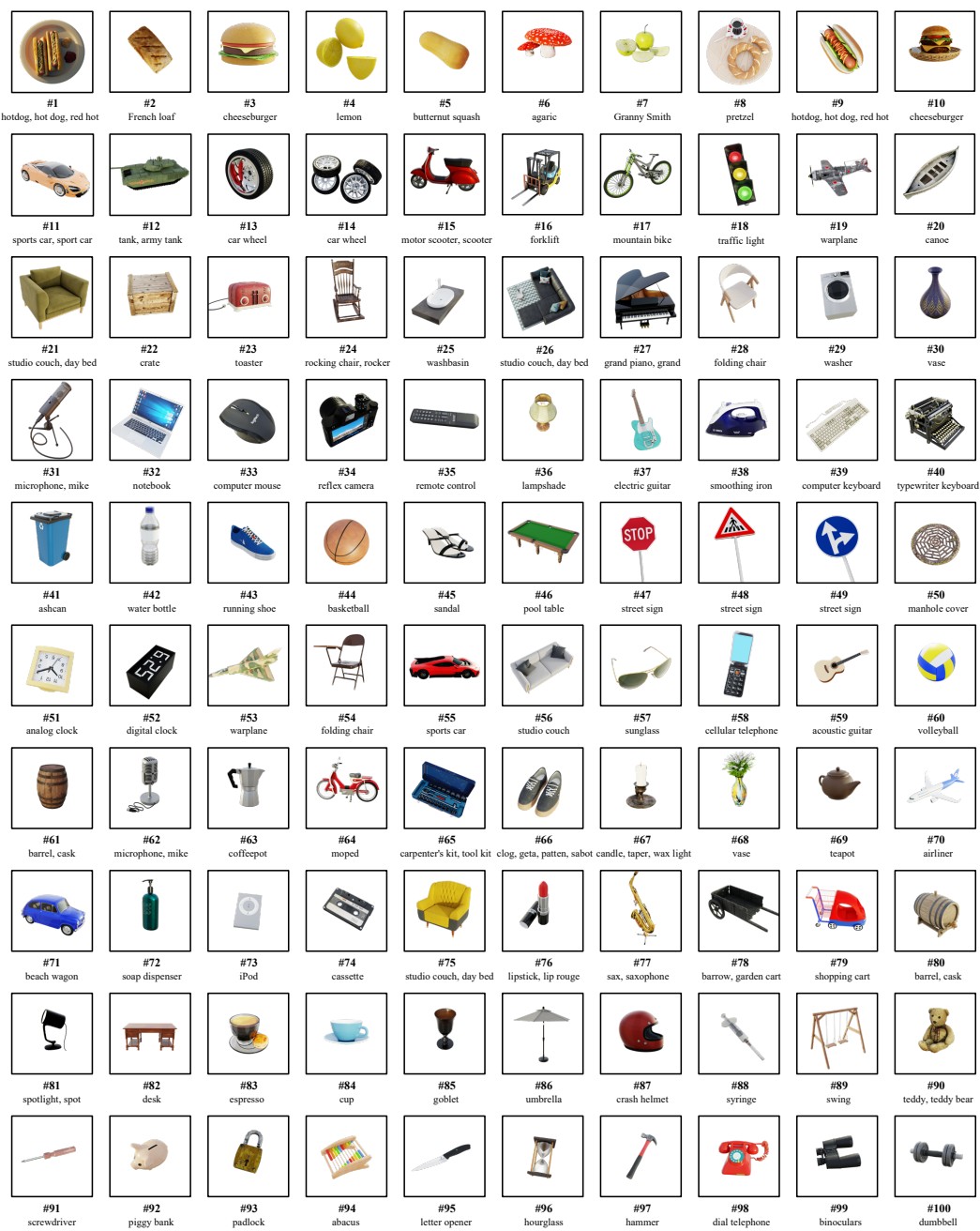

Figure B.1: Visualization of the 100 objects in our dataset.

## B  Dataset

In the experiments, we collect a dataset of 100 3D object models from BlenderKit. These objects were selected based on the following criteria. 1) They are common in the real world, including cars, street signs, etc; 2) They are easily recognizable by humans; and 3) They belong to the ImageNet classes such that the adopted visual recognition models (e.g., ResNet, ViT) can classify them from natural viewpoints with high accuracy. The royalty free license (https://www.blenderkit.com/docs/licenses/) states that: "This license protects the work in the way that it allows commercial use without mentioning the author, but doesn't allow for re-sale of the asset in the same form (eg. a

3D model sold as a 3D model or part of assetpack or game level on a marketplace)". As this work is not for commercial use, we do not violate the license.

Fig. B.1 visualizes these objects from natural viewpoints. We also show their corresponding ground-truth labels belonging to the 1000 ImageNet classes. A limitation of the dataset is the smaller size. It is because training NeRF for each object is computationally expensive as discussed in Appendix C.1. The dataset does not contain all classes in ImageNet, especially those with deformable shapes (e.g., animals), which may potentially lead to biased evaluations. Nevertheless, we think that the dataset is highly valuable for benchmarking the viewpoint robustness of visual recognition models, since it is important to understand model vulnerabilities to viewpoint changes in safety-critical applications while few efforts have been devoted to this area. We will continuously enlarge the dataset in the future.

# C    Additional experiments

We provide the additional experimental results in this section.

## C.1    Computation complexity

All of the experiments were conducted on NVIDIA GeForce RTX 3090 GPUs. For each object, it takes about 6.5 hours to train the NeRF model and about 4.5 hours to generate a distribution of adversarial viewpoints with a single GPU. So in total, running all of the experiments (including ablation studies and physical-world experiments) requires about 500 GPU days. A potential limitation of ViewFool is that it runs slowly due to the time-consuming rendering process of NeRF and query-based optimization algorithm. It can be further accelerated by adopting advanced NeRF variants with faster inference [15, 34], which we leave to future work. And therefore we prefer to establish a benchmark for viewpoint robustness evaluation of more classifiers rather than running ViewFool for each of them.

## C.2    NeRF results

The first step of our algorithm is to train a NeRF model for each object. The quality of the NeRF rendering also affects the performance of ViewFool since a smaller reality gap between the rendered images and the real images can make the generated adversarial viewpoints more robust in the real world. We provide the results of NeRF in Table C.1. The training images for NeRF are submitted in the code in the supplementary material.

## C.3    Visualization of the optimal adversarial viewpoints

To supplement the visualization results in Fig. 3, we provide the visualization of the optimal adversarial viewpoints $\mathbf{v}^*$ of all 100 objects in our dataset. Fig. C.1 shows the visualization results for all objects with the predicted labels and confidences. These images from adversarial viewpoints can be easily recognized by humans and look natural, which identifies the vulnerability of visual recognition models to natural changes of the inputs. From the figure, we can observe that some of the objects (e.g., #1, #9, #41) are flipped over, while some others (e.g., #3, #11, #19) are viewed from the above (i.e., bird's eye view). Since theses views are uncommon in the training datasets (e.g., ImageNet), the model is not endowed with the robustness. However, these strange viewpoints can occur in the real world, which can lead to severe security/safety problems.

## C.4    Diversity of ImageNet-V

ImageNet-V consists of 10000 images of 100 objects, in which we adopt 100 images from varying viewpoints sampled from the adversarial distributions. As shown in Table 2, we learn a wide range of viewpoints by ViewFool, such that the images in ImageNet-V do not overlap with each other and have diversity. To further demonstrate this, Fig. C.2 shows some images in ImageNet-V. We can observe that the sampled images for the same object are also different with each other, thus the diversity of ImageNet-V is improved.

Table C.1: The results of NeRF rendering.

| ID | Label | PSNR | ID | Label | PSNR |
|----|-------|------|----|-------|------|
| 1 | hotdog, hot dog, red hot | 35.50 | 51 | analog clock | 36.50 |
| 2 | French loaf | 34.80 | 52 | digital clock | 31.60 |
| 3 | cheeseburger | 35.80 | 53 | warplane | 35.90 |
| 4 | lemon | 34.80 | 54 | folding chair | 29.70 |
| 5 | butternut squash | 41.10 | 55 | sports car | 30.00 |
| 6 | agaric | 31.30 | 56 | studio couch | 40.00 |
| 7 | Granny Smith | 36.00 | 57 | sunglass | 34.30 |
| 8 | pretzel | 30.00 | 58 | cellular telephone | 32.20 |
| 9 | hotdog, hot dog, red hot | 28.60 | 59 | acoustic guitar | 32.50 |
| 10 | cheeseburger | 27.10 | 60 | volleyball | 35.10 |
| 11 | sports car | 27.00 | 61 | barrel, cask | 34.40 |
| 12 | tank | 34.60 | 62 | microphone | 32.40 |
| 13 | car wheel | 28.00 | 63 | moped | 28.60 |
| 14 | car wheel | 29.50 | 64 | carpenter's kit, tool kit | 28.30 |
| 15 | motor scooter, scooter | 31.90 | 65 | clog | 30.90 |
| 16 | forklift | 29.40 | 66 | candle | 32.20 |
| 17 | mountain bike | 26.40 | 67 | coffeepot | 35.10 |
| 18 | traffic light | 33.60 | 68 | vase | 27.10 |
| 19 | warplane | 30.10 | 69 | teapot | 36.90 |
| 20 | canoe | 31.50 | 70 | airliner | 38.40 |
| 21 | studio couch | 40.00 | 71 | beach wagon | 29.10 |
| 22 | crate | 33.10 | 72 | soap dispenser | 34.80 |
| 23 | toaster | 35.40 | 73 | iPod | 39.20 |
| 24 | rocking chair | 31.40 | 74 | cassette | 33.60 |
| 25 | washbasin | 33.80 | 75 | studio couch | 35.70 |
| 26 | studio couch | 31.00 | 76 | lipstick | 31.60 |
| 27 | grand piano | 27.90 | 77 | sax | 27.40 |
| 28 | folding chair | 35.80 | 78 | barrow, garden cart | 30.10 |
| 29 | washer | 36.50 | 79 | shopping cart | 29.10 |
| 30 | vase | 31.20 | 80 | barrel, cask | 33.50 |
| 31 | microphone | 28.40 | 81 | spotlight | 34.40 |
| 32 | notebook | 32.60 | 82 | desk | 35.10 |
| 33 | computer mouse | 35.50 | 83 | espresso | 26.70 |
| 34 | reflex camera | 30.00 | 84 | cup | 36.20 |
| 35 | remote control | 34.60 | 85 | goblet | 34.10 |
| 36 | lampshade | 33.50 | 86 | umbrella | 36.70 |
| 37 | electric guitar | 35.90 | 87 | crash helmet | 33.40 |
| 38 | smoothing iron | 31.40 | 88 | syringe | 36.10 |
| 39 | computer keyboard | 36.20 | 89 | swing | 34.40 |
| 40 | typewriter keyboard | 24.30 | 90 | teddy, teddy bear | 32.50 |
| 41 | ashcan | 33.50 | 91 | screwdriver | 39.30 |
| 42 | water bottle | 33.20 | 92 | piggy bank | 38.90 |
| 43 | running shoe | 33.70 | 93 | padlock | 35.90 |
| 44 | basketball | 33.40 | 94 | abacus | 36.50 |
| 45 | sandal | 28.60 | 95 | letter opener | 43.30 |
| 46 | pool table | 33.50 | 96 | hourglass | 32.40 |
| 47 | street sign | 36.60 | 97 | hammer | 40.00 |
| 48 | street sign | 33.60 | 98 | dial telephone | 30.80 |
| 49 | street sign | 34.20 | 99 | binoculars | 32.60 |
| 50 | manhole cover | 29.40 | 100 | dumbbell | 33.70 |

## C.5 Different ranges of rotation angles

We study the performance of using different ranges of rotation angles. Besides the baseline setting in Table 1 that $\psi \in [-180°, 180°]$, $\theta \in [-30°, 30°]$, $\phi \in [20°, 160°]$, we consider three settings that

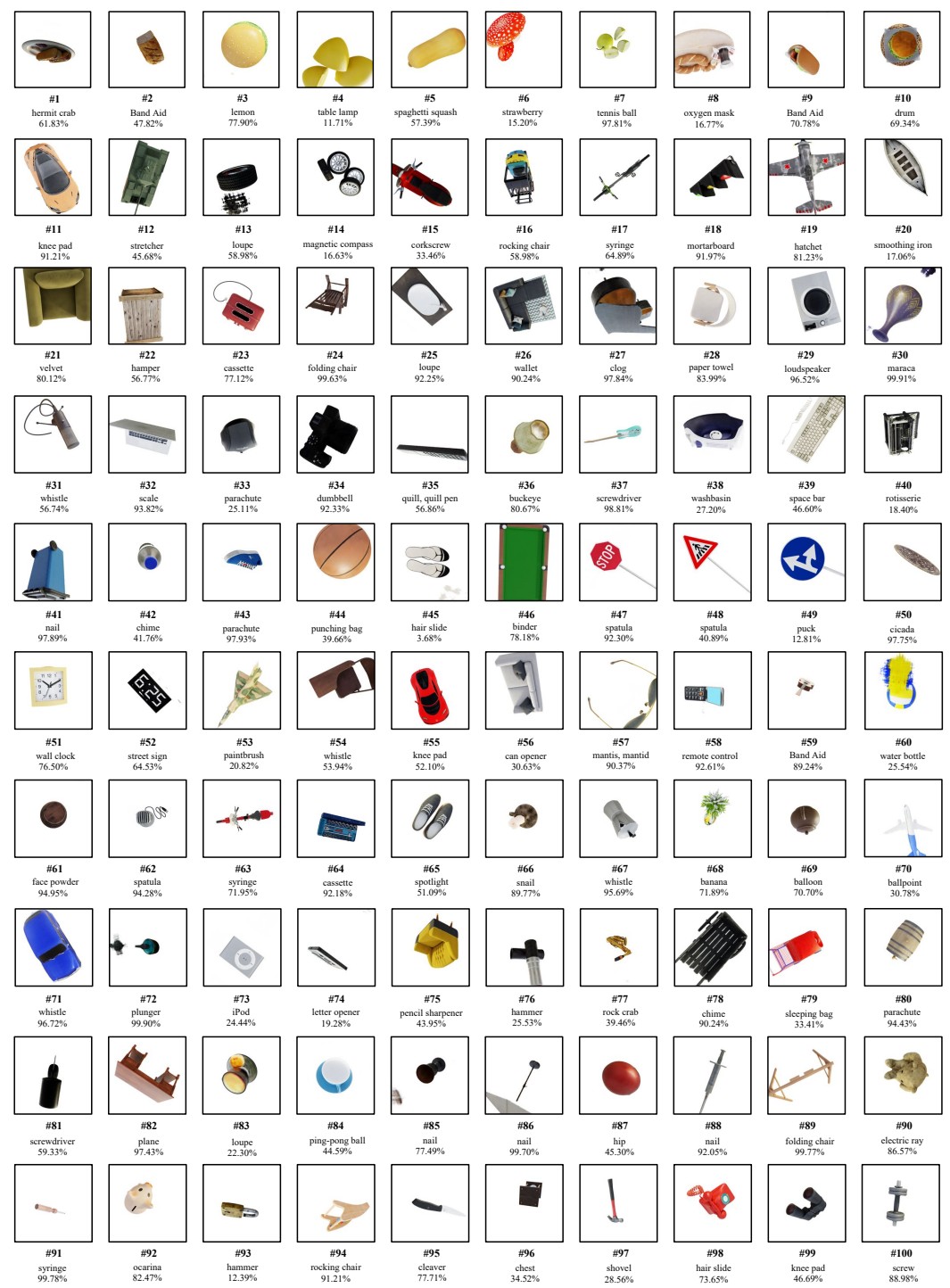

Figure C.1: Visualization of the 100 objects from the optimal adversarial viewpoints.

we only optimize one rotation angle while keep the other two rotation angles fixed. We also consider two more settings that we decrease the range of rotation angles. In this ablation study, we do not optimize the translation parameters. The results on ResNet-50 are shown in Table C.2. Generally, a larger range of rotation angles leads to better performance due to the larger search space.

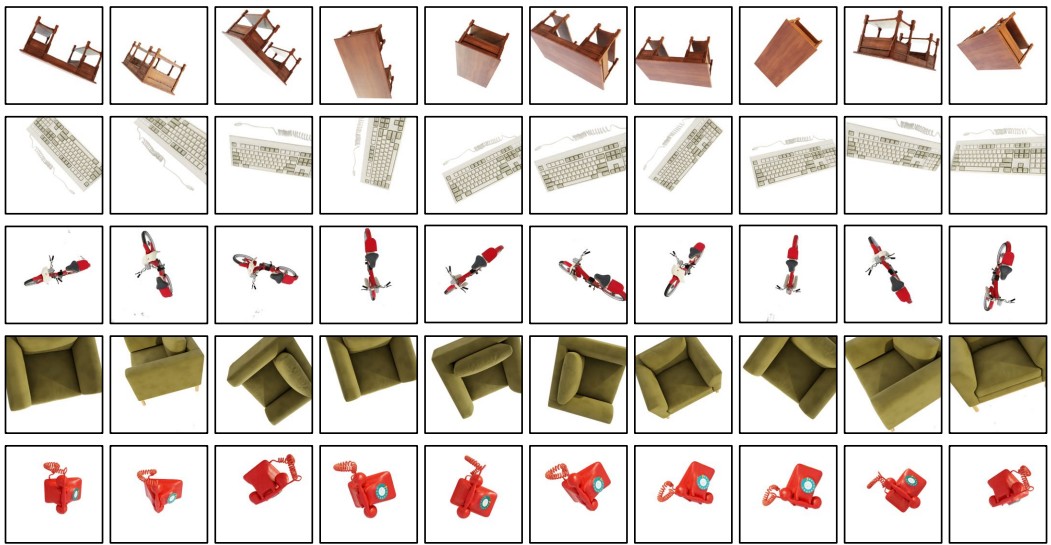

Figure C.2: Sampled images from the ImageNet-V dataset.

Table C.2: The results of different ranges of rotation angles.

| $\psi$ | $\theta$ | $\phi$ | $\mathcal{R}(p^*(\mathbf{v}))$ | $\mathcal{R}(\mathbf{v}^*)$ |
|---|---|---|---|---|
| $[-180°, 180°]$ | $0°$ | $90°$ | 66.46% | 77% |
| $0°$ | $[-30°, 30°]$ | $90°$ | 59.64% | 64% |
| $0°$ | $0°$ | $[20°, 160°]$ | 75.21% | 81% |
| $[-45°, 45°]$ | $[-7.5°, 7.5°]$ | $[72.5°, 107.5°]$ | 68.57% | 79% |
| $[-90°, 90°]$ | $[-15°, 15°]$ | $[55°, 125°]$ | 77.27% | 91% |
| $[-180°, 180°]$ | $[-30°, 30°]$ | $[20°, 160°]$ | **84.25%** | **96%** |

Table C.3: Comparison to adversarial 2D transformations.

| | $\mathcal{R}(p^*(\mathbf{v}))$ | $\mathcal{R}(\mathbf{v}^*)$ |
|---|---|---|
| 3D Viewpoints | **88.79%** | **98%** |
| 2D Transformations | 76.93% | 85% |

## C.6  Comparison to adversarial 2D transformations

We further study the performance of ViewFool compared to adversarial 2D transformations. Note that 2D transformations (including 2D rotation, scaling, cropping) are a subset of 3D viewpoint changes studied in this paper. Specifically, if we keep the rotation angles $\psi$ and $\phi$ fixed, and optimize other viewpoint parameters including $\theta$ and $[\Delta_x, \Delta_y, \Delta_z]$, the changes correspond to 2D image rotation, translation, scaling, and cropping. So in general, adversarial viewpoints can lead to a higher attack success rate than adversarial 2D transformations. To validate this, we conduct an ablation study by keeping $\psi = 0°$, $\phi = 90°$ fixed, and perform optimization over the other parameters using the same algorithm and experimental settings. The results on ResNet-50 are shown in Table C.3. It can be seen that 3D viewpoint changes lead to better attack success rates than 2D image transformations.

## C.7  Experiments on the Objectron dataset

We conduct experiments on the Objectron dataset [1], which contains object-centric videos in the wild. We select 10 videos that the objects can be correctly classified by the ResNet-50 model (otherwise it is meaningless to study viewpoint robustness). We adopt the same experimental settings to train NeRF models and generate adversarial viewpoints for those objects. Fig. C.3 shows the visualization of the adversarial viewpoints against ResNet-50. ViewFool successfully generates adversarial viewpoints

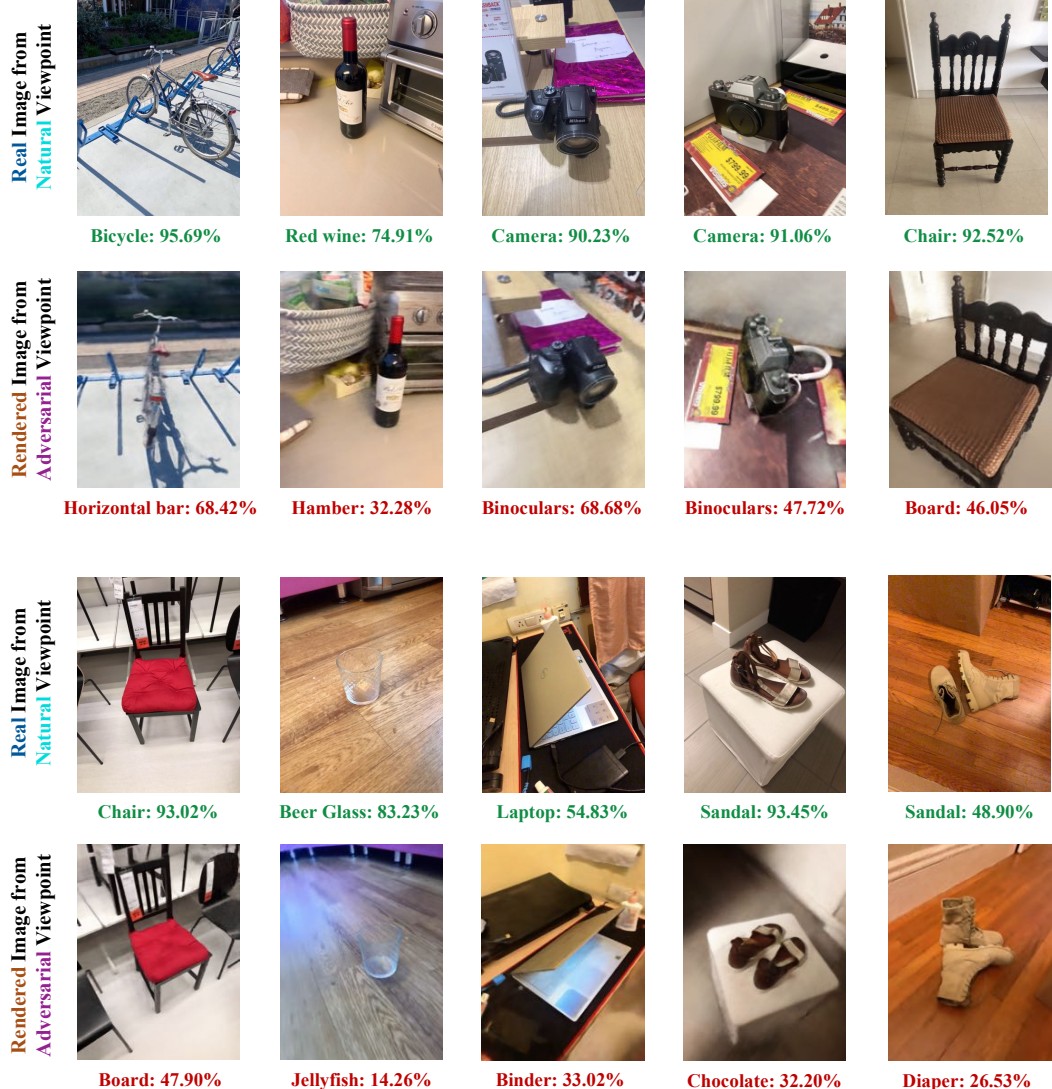

Figure C.3: Visualization of the adversarial viewpoints generated by ViewFool against ResNet-50 on the Objectron dataset. The first and third rows show the real images taken from natural viewpoints that can be correctly classified. The second and fourth rows show the rendered images from adversarial viewpoints $\mathbf{v}^*$.

for all objects (i.e., the attack success rate is $100\%$). However, since we do not have the ground-truth 3D models of objects/scenes, we cannot obtain the real images taken from the adversarial viewpoints.

## C.8  More real-world experiments

We further conduct real-world experiments on another 4 objects, including two indoor objects (chair and keyboard) and two outdoor objects (street sign and traffic light). In this experiment, we do not place white paper underneath the object to be more realistic in the wild. Besides Fig. 1, Fig. C.4 shows the visualization results of rendered images and more captured images from adversarial viewpoints. ViewFool successfully generates adversarial viewpoints for all these 4 objects in the real world.

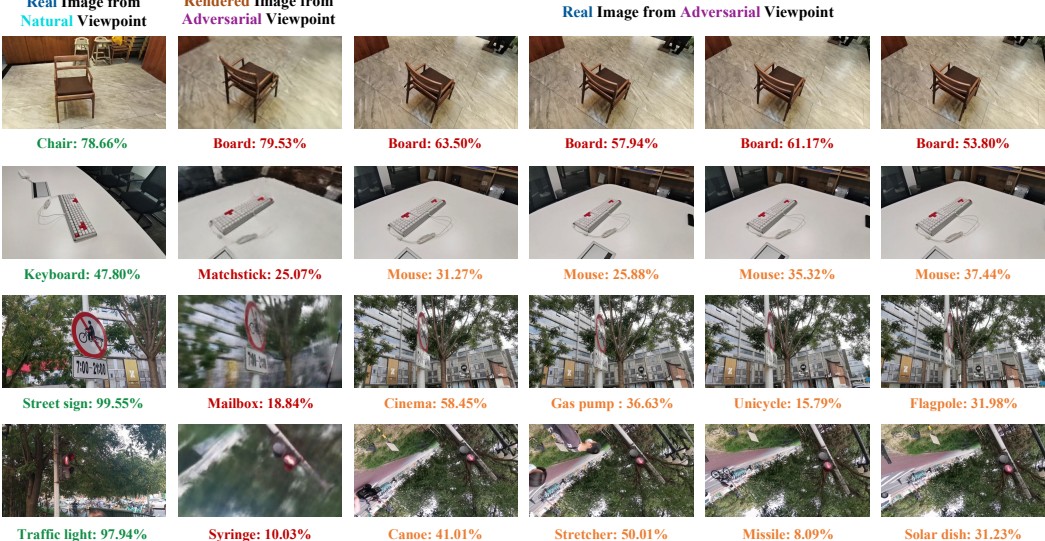

Figure C.4: Visualization of another 4 real-world objects in the wild given different viewpoints. The first column shows the real images from natural viewpoints. The second column shows the rendered images from adversarial viewpoints. The 3-7 columns show the real images taken to approximate the adversarial viewpoints.