# OpenReview forum: "ViewFool: Evaluating the Robustness of Visual Recognition to Adversarial Viewpoints"
_NeurIPS.cc/2022/Conference — NeurIPS 2022 Accept_

### Official Review · Reviewer_RtMV · 2022-06-24

**Rating:** 5
**Confidence:** 5
**Soundness:** 4 excellent
**Presentation:** 3 good
**Contribution:** 3 good

**Summary:**

This paper studies the viewpoint robustness of trained visual recognition models. People have studied adversarial perturbations on 2D images for a long time, but not all of them are physically plausible. There were some previous efforts that tried to generate adversarial examples in the 3D physical space, but they used synthetic objects and rendering software. This paper proposes to replace these with NeRF, so that it becomes possible to conduct the same study on real-world objects. The authors conducted extensive experiments and demonstrated that trained visual recognition models can still be very vulnerable to simple viewpoint changes.

POST-REBUTTAL UPDATE:

Make no mistake: I really like and enjoy the paper. However, I am also concerned by the fact that the authors used models trained on ImageNet in their experiments. The big domain gap makes the random attack also very effective, and it is harder to put the proposed approach in perspective. At the very least, I was hoping the authors could train a model on their dataset, and put that in addition to the ImageNet trained one they have been using. In light of this, I have decided to slightly decrease my score from 6 to 5, which is still a positive score.

**Questions:**

1. In Introduction (say around L39), the authors criticized that previous papers / methods rely on synthetic 3D object models, implying that they are different. However, other than the experiments in Section 4.3, a lot of experiments (say those in Section 4.1) are using BlenderKit, which is still 3D models. Therefore I feel that the writing ought to be improved, otherwise the difference w.r.t. previous papers / methods (say [61]) could be perceived as exaggerated / overrated.

2. I am a bit concerned that the random search baseline in Table 1 also attack the model pretty successfully. By contrast, random search within the pixel RGB neighborhood on 2D images have a very low attack success rate. To me, this raises questions about the quality of the trained model. Maybe sampling 100 images per 3D object on the upper hemisphere is not enough. Maybe the training protocol is not well-tuned. My point is, if a model is full of weaknesses, then we were not going to deploy the model anyway, and the significance of studying its adversarial robustness becomes much smaller. Have you considered training a stronger model that cannot be easily attacked by random search? And is it possible to conduct the experiment on a state-of-the-art published and released checkpoint (so that the quality is guaranteed)?

**Limitations:**

The authors discussed some limitations in the Conclusion section, but I think they could be expanded. For example, the authors could have discussed whether or not the trained models are strong enough on these datasets newly proposed in this work. Another limitation worth discussing is that I feel the current work still largely fall into the "synthetic 3D object" category (e.g. white paper is added etc), and the NeRF-generated images here are still more like ShapeNet than ImageNet (or Block-NeRF for that matter).

**Strengths And Weaknesses:**

Strengths:
- I believe the problem being studied is an important one. We need to understand how models trained on selective examples (views) generalize to the generic setting.
- I especially like the experiments in Section 4.3, where the authors showed that it is possible to reproduce the adversarial viewpoints found by NeRF in the real world, and the attack can still be successful.
- Experiments are very extensive, covering a lot of different axes (architecture, with or without pre-training, various datasets).

Weaknesses:
- I have some concerns regarding presentation and baseline. See the questions section below.

Overall I think the originality, quality, clarity, significance of this paper are all above average.

---

> ### Author Response · Authors · 2022-08-02
> **Thank you for the valuable review**
>
> Thank you for appreciating our new contributions as well as providing the valuable feedback. We have uploaded a revision of our paper. Below we address the detailed comments and hope that you may find our response satisfactory.
>
> ***Question 1: The use of synthetic 3D object models in previous papers and our paper.***
>
> Thank you for pointing this out. We want to clarify that our method and previous ones adopt synthetic 3D object models for different purposes. The previous methods directly render the synthetic 3D objects into images to perform attacks. They can hardly be used to generate adversarial viewpoints for real-world objects without accurate 3D models. Differently, our method adopts NeRF to learn implicit representations of 3D objects given multi-view images. Thus our method can be deployed for real-world objects, as shown in Section 4.3. We adopted the synthetic 3D objects in the experiments mainly for the reproducibility purpose -- although we have the ground-truth 3D objects, we still take multi-view images as inputs to our algorithm. In the revision, we revise our argument in the Introduction to make it not exaggerated / overrated.
>
> ***Question 2: A concern about the random search baseline.***
>
> Actually, the image classification models ResNet-50 and ViT-B/16 are not trained on the 100 images of each 3D object. They are public models from the `timm` library (https://github.com/rwightman/pytorch-image-models) that are trained on the ImageNet dataset. The 40 image classifiers used for the ImageNet-V benchmark in Section 4.4 are also public models.
> The high success rate of the random search baseline is mainly caused by the large range of viewpoint parameters. Since the classifiers are trained on the standard ImageNet dataset with most images taken from natural viewpoints, the images from (randomly sampled) novel viewpoints can lead to a relatively high attack success rate (~50\%).
> To further show the effectiveness of ViewFool compared to the random search baseline, we restrict the range of viewpoint parameters to be smaller. For example, by setting the range of rotation as $\psi\in[-45^{\circ}, 45^{\circ}]$, $\theta\in[-7.5^{\circ}, 7.5^{\circ}]$, $\phi\in[72.5^{\circ}, 107.5^{\circ}]$, the attack success rate of the random search baseline is 27\%, while our method achieves 79\% success rate, which is much higher.
>
> ***Question 3: Limitations can be expanded.***
>
> As clarified in the above question, the image classifiers are all public models with good performance on ImageNet. For the limitation of using synthetic 3D object models, we further conduct experiment on the Objectron dataset [1], which contains object-centric videos in the wild. The experiments in Appendix C.7 demonstrate that our method can be reliably used to generate adversarial viewpoints for real-world datasets.
> We made a great effort to address most of the limitations, trying to make this work more solid and significant. The remaining limitations of this work include high computational cost of the proposed method, small dataset size used in the experiments, and no defense techniques being proposed. Nevertheless, they do not affect the main contributions of this work. In the revision, we list these limitations in the Conclusion and hope to address them in future work.
>
> **Reference:**
>
> [1] Ahmadyan et al. Objectron: A Large Scale Dataset of Object-Centric Videos in the Wild with Pose Annotations. CVPR 2021.

---

### Official Review · Reviewer_44qA · 2022-07-08

**Rating:** 6
**Confidence:** 4
**Soundness:** 3 good
**Presentation:** 3 good
**Contribution:** 3 good

**Summary:**

The paper proposes a technique to generate adversarial viewpoints of multi-view objects for training image classification models. It uses Neural Rendering as an implicit representation to differentiably optimize the viewpoint parameters to synthesize adversarial images. It introduces an entropic regularizer for optimizing a parameterized distribution which leads to higher diversity of content.

**Questions:**

1. The maximization function could lead to generation of highly skewed viewpoints. However, looking at the maximum likelihood estimates (v*), it doesn't seem to be the case. How do you constrain the viewpoints to remain within reasonable limits?

2. The gap between the Real(v*) and Rendered(v*) is significant. Why is rendered(v*) always harder to classify than the real(v*). Is it because of some artifacts introduced by the rendering technique. Can you perform some ablation studies to explain this gap?

3. It is not clear if the NeRF weights are updated during training.
What is the effect on the NeRF renderer after the adversarial optimization is performed.
A comparison between the baseline PSNR (that's given in the appendix) and post-optimization PSNR could be a useful datapoint to see if the NeRF degrades in fidelity.


**Limitations:**

The limitations given in the appendix reasonably capture the issues with the proposed dataset and the baseline NeRF model. The authors should ponder over the limitations of the proposed technique. Perhaps the questions raised in the previous two sections can help guide the authors to find more relevant points.

**Strengths And Weaknesses:**

Strengths:
- The paper shows that Neural Rendering can be used as an implicit representation to create adversarial viewpoints for image classification.
- A forward propagation technique based on Natural Evolution Strategies is proposed to bypass the expensive NeRF rendering process during optimization.

Weaknesses:
- The optimization problem is posed as an unbounded maximization function on classification loss and distribution entropy. It is not clear to me if it would work for general, real-world images where the model could diverge to nonsensical neural rendering.

- The results are shown on simple images with white background. Even the real-world dataset used in the paper is not representative of actual image classification datasets such as ImageNet. It is not clear if the method would generalize to real-world images. Recent datasets such as CO3D or Objectron could be easily used to demonstrate the efficacy of the method.

- The method presupposes availability of multi-view images of objects to train the underlying NeRF. It would not work for single shot images of objects.

- Lacks comparison to other SoTA approaches for adversarial generation for training.

---

> ### Author Response · Authors · 2022-08-02
> **Thank you for the valuable review**
>
> Thank you for the valuable review. We have uploaded a revision of our paper. Below we address the detailed comments, and hope that you may find our response satisfactory.
>
> ***Question 1: Does the  unbounded optimization problem work for general, real-world images?***
>
> Actually, the proposed method optimizes the viewpoint parameters given a well-trained NeRF model.
> Therefore, if the training of NeRF can converge for real-world images, our method can be applied consequently. In general, NeRF and its recent extensions work well for modeling real-world objects, thus our method can also generate adversarial viewpoints for general, real-world images.
>
> ***Question 2: Demonstrate the efficacy of the method with the recent datasets.***
>
> Thanks for the suggestion. We further conduct experiments on the Objectron dataset [1], which contains object-centric videos in the wild. We select 10 videos that the objects can be correctly classified by ResNet-50. As shown in Figure C.3 in Appendix C.7, we successfully generate adversarial viewpoints for all objects (i.e., 100% attack success rate), showing the efficacy of our method on real-world images. Besides, we further conduct real-world experiments on another 4 objects, including two indoor objects (chair and keyboard) and two outdoor objects (street sign and traffic light). Our method also succeeds to generate adversarial viewpoints for all objects, as shown in Figure C.4 in Appendix C.8. We will provide more comprehensive evaluations in the final.
>
> ***Question 3: The method presupposes availability of multi-view images of objects.***
>
> Our method generates adversarial viewpoints based on a neural rendering model that can synthesize photo-realistic images from novel viewpoints. We adopted the vanilla NeRF in this paper to demonstrate the feasibility and effectiveness of our method, but it is also compatible with other neural rendering methods. Since the original NeRF needs multi-view images of each object, our method inevitably has the same assumption. However, as more advanced NeRF-variants (e.g., pixelNeRF [2]) have been proposed to use one or few images of the object, our method can also be extended to this setting by using these NeRF-variants.
>
> ***Question 4: Lacks comparison to other SoTA approaches.***
>
> In this paper, we study a new problem that aims to generate adversarial viewpoints based on NeRF and there exists no prior work on generating adversarial viewpoints of real-world objects (as agreed by the other reviewers). Thus we compared with a random search baseline and a variant of our method without the entropic regularizer. For adversarial defenses, we actually adopted the SoTA adversarially trained models on ImageNet from [3]. As shown in Figure 5, the adversarially trained models have poor viewpoint robustness.
>
> ***Question 5: How do you constrain the viewpoints to remain within reasonable limits?***
>
> In fact, the viewpoint parameters $\mathbf{v}$ are bounded to avoid highly skewed viewpoints, as introduced in Line 132. We adopt the transformation of variable approach in Eq. (3) to make the optimization problem unbounded, such that it can be solved more effectively.
>
> ***Question 6: Why is rendered($v^{\ast}$) always harder to classify than the real($v^{\ast}$)?***
>
> This is because we generated the adversarial viewpoint based on the rendered image $\mathcal{R}(\mathbf{v})$ as shown in Eq. (2). Thus the rendered image from the adversarial viewpoint is more likely to fool the model. As discussed in Line 150-158, since there inevitably exists the artifact in the rendered image, the real image $\mathrm{Real}(\mathbf{v}^*)$ from the adversarial viewpoint may not fool the model due to the appearance difference. Our proposed method ViewFool adopts an entropic regularizer to reduce the gap between the performance of the rendered and real images, such that ViewFool with $\lambda=0.01$ has a smaller gap than ViewFool with $\lambda=0$, as discussed in Line 237-240. Figure 2 (right) shows an example to reflect this gap.
>
> ***Question 7: It is not clear if the NeRF weights are updated during training.***
>
> The weights of NeRF are **not** updated. In general, we first train the NeRF model given multi-view images of the object, then run ViewFool to generate adversarial viewpoints given the trained NeRF. In the revision, we clarify this in Line 126. For the concern about the NeRF renderer after the adversarial optimization, the average PSNR of all objects from natural viewpoints is 33.14, while that from adversarial viewpoints is 32.25. It can be seen that the quality of the rendered images is not degraded. Thus the viewpoint change is the main cause of model failures.
>
> **Reference:**
>
> [1] Ahmadyan et al. Objectron: A Large Scale Dataset of Object-Centric Videos in the Wild with Pose Annotations. CVPR 2021.
>
> [2] Yu et al. pixelNeRF: Neural Radiance Fields from One or Few Images. CVPR 2021.
>
> [3] Salman et al. Do Adversarially Robust ImageNet Models Transfer Better? NeurIPS 2020.

---

> > ### Author Response · Authors · 2022-08-07
> > **Look forward to further feedback**
> >
> > Dear Reviewer 44qA,
> >
> > We thank you again for the valuable comments. We hope you might find the response satisfactory and are looking forward to hearing from you about any further feedback.
> >
> > Best, Authors

---

> > > ### Comment · Reviewer_44qA · 2022-08-08
> > > **Thanks for the detailed rebuttal**
> > >
> > > Dear authors,
> > >
> > > Thank you for responding with additional details. The new results with the Objectron dataset indeed helps in proving the feasibility of the approach. The explanation on unbounded optimization is reasonable, but might require actual implementation to clarify remaining doubts. The authors have done a good job with clarifying the questions from me and the other reviewers, and I am happy to raise my score from 4 to 6.

---

> > > > ### Author Response · Authors · 2022-08-09
> > > > **Thanks for the update**
> > > >
> > > > Thank you very much for the increase on score and valuable feedback. We will provide further clarification on the unbounded optimization and improve the paper in the final.

---

### Official Review · Reviewer_ne6h · 2022-07-10

**Rating:** 6
**Confidence:** 3
**Ethics Flag:** Yes
**Soundness:** 3 good
**Presentation:** 3 good
**Contribution:** 3 good

**Summary:**

The authors propose to study viewpoint robustness of image classification models using 3D objects generated from different viewpoints. A new method to find adversarial viewpoints based on NeRF is proposed, and used to create a benchmark (ImageNet-V) for evaluating viewpoint robustness for image classification. The paper evaluates a set of image classifiers covering standard architectures on ImageNet-V, training objectives, and data augmentation strategies, which finding that it is a challenging dataset compared to using natural views only.

**Questions:**

I listed some questions above in weaknesses.

Some clarification questions:
- How dependent are the adversarial viewpoint / classification results on the NeRF training?

- In Figure 4, real-images taken from adversarial viewpoint, why is the lightning so different than the original (real from natural viewpoint)?

- How are the models in BlenderKit generated? Are they real or synthesized objects?


**Ethics Review Area:**

["Legal Compliance (e.g., GDPR, copyright, terms of use)"]

**Limitations:**

Yes the authors included a discussion on limitations in Appendix C.

However, I'm not familiar with Royalty Free licensing from BlenderKit (which this benchmark uses assets from) and whether images using these models can be used in a paper/as a benchmark.

**Strengths And Weaknesses:**

Strengths
- To the best of my knowledge, generating adversarial viewpoints of real objects using NeRF is new. The advantages mentioned by the authors in lines 147 to 160 looks reasonable.

- The authors perform fairly extensive evaluations on their proposed benchmark for evaluating viewpoint robustness (ImageNet-V), covering standard architectures, training objectives, and data augmentations.

- The paper is well-written with clear figures.

Weaknesses
- Using only 100 objects for benchmarking seems to be on the smaller side to me - maybe the authors can discuss more on the limitations of potential bias in this dataset due to the smaller amount of objects? Seems from the Appendix that all models are from BlenderKit - how were the objects selected?

- How does the 3D viewpoint change studied here compare to simple adversarial 2D transformations (rotation, scaling, cropping)? For example, the street-sign in Figure 2 looks simply rotated. The experiments would be stronger with some comparisons in this direction.

- The results in Figure 5 is not really significant without Figure C3 (which is in Appendix). My understanding is that Figure C3 is the performance under natural viewpoints & Figure 5 is with viewpoint robustness - I recommend that the authors present the bars in C3 with Figure 5, otherwise it is difficult to know how much the performance of each model drops.

- I feel the "in the Wild" phrase in the title is not really an accurate representation of this work - I would recommend that the authors reconsider this framing (perhaps "Adversarial" should be somewhere in the title instead). The benchmark consists of images from generated view points with a white background (ex: samples in Figure C2), which doesn't look "in the wild" - as a result, I'm not sure if this title is accurate.

---

> ### Author Response · Authors · 2022-08-02
> **Thank you for the valuable review (Part 2/2)**
>
> ***Question 5: How dependent are the adversarial viewpoint / classification results on the NeRF training?***
>
> Since the optimization problem (2) involves the rendering process of NeRF, our proposed method ViewFool is somewhat dependent on the performance of NeRF. As discussed in Line 150-158, if the rendered images have significant appearance differences to the real images, the generated adversarial viewpoints may not fool the model given real objects. ViewFool addresses this issue by learning the distribution of adversarial viewpoints to mitigate the gap between neural rendering and real objects. The results in Table 1 validate that the performance gap between the real images and rendered images from the adversarial viewpoints are small.
>
> ***Question 6: Why is the lighting different in Figure 4?***
>
> This is because the images are taken in different days. In this experiment, we first took a video of each object for training NeRF. The first column of Figure 4 shows the real image sampled from this video. We then trained a NeRF for each object and ran ViewFool to generate the adversarial viewpoints. This procedure needed about one day to complete. After that, we re-took photos (shown in the 3-7 columns in Figure 4) of the object from the adversarial viewpoints in the next day that the lighting conditions changed over time. Although the lighting has changed, the generated adversarial viewpoints remain effective to fool the model. A potential improvement of our method is to consider relighting techniques in NeRF.
>
> ***Question 7: How are the models in BlenderKit generated?***
>
> The models in BlenderKit are synthetic objects. As shown in Figure B.1 in Appendix B, we manually selected these objects that are natural and common in the real world.
>
> **Reference:**
>
> [1] Ahmadyan et al. Objectron: A Large Scale Dataset of Object-Centric Videos in the Wild with Pose Annotations. CVPR 2021.

---

> > ### Comment · Reviewer_ne6h · 2022-08-06
> > **Updated Rating**
> >
> > I appreciate the authors for their response and the new experiments addresses most of my concerns - I've updated my rating from 5 to 6.

---

> > > ### Author Response · Authors · 2022-08-07
> > > **Thanks for the update**
> > >
> > > Thank you very much for the increase on rating and valuable feedback. We highly appreciate that. We'll try our best to further improve in the final version.

---

> ### Author Response · Authors · 2022-08-02
> **Thank you for the valuable review (Part 1/2)**
>
> Thank you for appreciating our new contributions as well as providing the valuable feedback. We have uploaded a revision of our paper. Below we address the detailed comments, and hope that you may find our response satisfactory.
>
> ***Question 1: Discuss more on the limitations of potential bias in this dataset due to the smaller amount of objects. How were the objects selected?***
>
> As shown in Figure B.1 in Appendix B, the 100 objects were manually selected based on the following criteria. 1) They are common in the real world, including cars, street signs, etc; 2) They are easily recognizable by humans; and 3) They belong to the ImageNet classes such that the adopted visual recognition models (e.g., ResNet, ViT) can classify them from natural viewpoints with high accuracy.
> Since training NeRF for each object is computationally expensive as discussed in Appendix C.1, the number of objects in the dataset is relatively small. Thus the dataset does not contain all classes in ImageNet such as animals, which is a potential limitation of biased classes. In the revision, we make this clearer in Appendix B. Nevertheless, we think that the dataset is highly valuable for benchmarking the viewpoint robustness of visual recognition models, since it is important to understand model vulnerabilities to viewpoint changes in safety-critical applications while few efforts have been devoted to this area. It can also facilitate future research on improving viewpoint robustness. We will continuously enlarge the dataset in the future.
>
> ***Question 2: How does the 3D viewpoint change compare to adversarial 2D transformations?***
>
> Thanks for the suggestion. Note that 2D transformations (including 2D rotation, scaling, cropping) are a subset of 3D viewpoint changes studied in this paper. Specifically, if we keep the rotation angles $\psi$ and $\phi$ fixed, and optimize other viewpoint parameters including $\theta$ and $[\Delta_x, \Delta_y, \Delta_z]$, the changes correspond to 2D image rotation, translation, scaling, and cropping. So in general, adversarial viewpoints can lead to a higher attack success rate than adversarial 2D transformations. To validate this, we conduct an ablation study by keeping $\psi=0^{\circ}$, $\phi=90^{\circ}$ fixed, and perform optimization over the other parameters using the same algorithm and experimental settings. The results on ResNet-50 are shown below.
>
> | | $\mathcal{R}(p^*(\mathbf{v}))$ | $\mathcal{R}(\mathbf{v}^*)$ |
> | :----- | :-----: | :----: |
> | 3D Viewpoints | **88.79\%** | **98\%** |
> | 2D Transformations | 76.93\% | 85\% |
>
> It can be seen that 3D viewpoint changes lead to higher attack success rates than 2D image transformations. In the revision, we add the results in Appendix C.6.
>
> ***Question 3: Figure 5 is not significant without Figure C3.***
>
> Thanks for the suggestion. In the revision, we show the model performance under natural viewpoints and on ImageNet-V in Figure 5 for better comparison.
>
> ***Question 4: Is "in the wild" in the title correct?***
>
> We adopted the phrase "in the wild" to indicate that the proposed method can be deployed in the real world to generate adversarial viewpoints of physical objects. However, to conduct fairer and more reproducible experiments, we adopted synthetic 3D objects with the white background. Our method is also applicable to real-world datasets. In the revision, we provide the results on the Objectron dataset [1] in Appendix C.7, which contains object-centric videos in
> the wild. To avoid misunderstanding and make the title more accurate, we change it to "ViewFool: Evaluating the Robustness of Visual Recognition to Adversarial Viewpoints" (it seems that the title on OpenReview cannot be changed at this time, so we will update the title in the final).

---

### Official Review · Reviewer_9ojy · 2022-07-11

**Rating:** 7
**Confidence:** 4
**Soundness:** 4 excellent
**Presentation:** 2 fair
**Contribution:** 4 excellent

**Summary:**

The submission proposes a novel task related to NeRF: finding adversarial viewpoints that mislead visual recognition models. When searching adversarial viewpoints, they find search viewpoint distribution rather than one discrete viewpoint and suggest a novel optimization algorithm that fits their objective.

**Questions:**

Is it critical to the performance to use other ranges of the rotation angles? If so, please report the ablation of that.

**Limitations:**

This paper is limited to background-free or simple background images, which has room for improvement to use more realistic and complicated datasets such as the dataset suggested by MipNeRF-360.

**Strengths And Weaknesses:**

Strengths:

- It is the first work to tackle searching adversarial viewpoint on the NeRF-related methodologies.
- They suggest several effective and reasonable methods, searching the viewpoints via distribution, reducing memory requirements for computing gradients based on NES, and adopting reparameterization tricks to reduce the variance for computing gradients.
- The experiments are solid.
- It also shows the resistance to camera fluctuations, demonstrating its effectiveness in real-world scenarios.

Weaknesses:

- Their experiments are limited to background-free or simple background images

---

> ### Author Response · Authors · 2022-08-02
> **Thank you for the valuable review**
>
> Thank you for appreciating our new contributions as well as providing the valuable feedback. We have uploaded a revision of our paper. Below we address the detailed comments.
>
> ***Question 1: The experiments are limited to background-free or simple background images.***
>
> We adopted background-free images mainly for evaluating the viewpoint robustness of visual recognition models without the interference of background variations. Our method is generally applicable for more realistic datasets. To demonstrate this, we further conduct experiments on the Objectron dataset [1], which contains object-centric videos in the wild. As shown in Figure C.3 in Appendix C.7, we successfully generate adversarial viewpoints for all selected objects in this dataset.
>
> ***Question 2: The performance of using other ranges of the rotation angles.***
>
> In the revision, we conduct an additional ablation study of using different ranges of rotation angles. Besides the baseline setting in Table 1 that $\psi\in[-180^{\circ},180^{\circ}]$, $\theta\in[-30^{\circ}, 30^{\circ}]$, $\phi\in[20^{\circ}, 160^{\circ}]$, we include three settings that we only optimize one rotation angle while keeping the other two rotation angles fixed. We also consider two more settings that we decrease the range of rotation angles. The results on ResNet-50 are shown below and in Table C.2 in Appendix C.5.
>
> | $\psi$ | $\theta$ | $\phi$ | $\mathcal{R}(p^*(\mathbf{v}))$ | $\mathcal{R}(\mathbf{v}^*)$ |
> | :-----: | :-----: | :----: | :-----: | :-----: |
> | $[-180^{\circ}, 180^{\circ}]$ | $0^{\circ}$ | $90^{\circ}$ | 66.46\% | 77\% |
> | $0^{\circ}$ | $[-30^{\circ}, 30^{\circ}]$ | $90^{\circ}$ | 59.64\% | 64\% |
> | $0^{\circ}$ | $0^{\circ}$ | $[20^{\circ}, 160^{\circ}]$ | 75.21\% | 81\% |
> | $[-45^{\circ}, 45^{\circ}]$ | $[-7.5^{\circ}, 7.5^{\circ}]$ | $[72.5^{\circ}, 107.5^{\circ}]$ | 68.67\% | 79\% |
> | $[-90^{\circ}, 90^{\circ}]$ | $[-15^{\circ}, 15^{\circ}]$ | $[55^{\circ}, 125^{\circ}]$ | 77.27\% | 91\% |
> | $[-180^{\circ}, 180^{\circ}]$ | $[-30^{\circ}, 30^{\circ}]$ | $[20^{\circ}, 160^{\circ}]$ | **84.25\%** | **96\%** |
>
> The attack success rate is indeed affected by the range of rotation angles. Generally, a larger range of rotation angles leads to better performance due to the larger search space.
>
> **Reference:**
>
> [1] Ahmadyan et al. Objectron: A Large Scale Dataset of Object-Centric Videos in the Wild with Pose Annotations. CVPR 2021.

---

### Review · Ethics_Reviewer_pW6V · 2022-08-08

**Recommendation:** All ethics concerns have been addressed.

**Ethics Review:**

This paper presents a new method using NeRF to generate adversarial viewpoints of real objects to evaluate viewpoint robustness of image classification. A potential ethics concern with work is that the work also reveals the vulnerability of image classifications to viewpoint changes, and invites adversarial attacks (potentially on real world systems). The authors did acknowledge this issue in the paper. At the same time, the work could also stimulate future research on defense mechanisms to viewpoint changes.

As for legal compliance, copyrights and data usage, the authors clarified in the appendix B that the images used in the paper are royalty free for non-commercial use.

---

### Author Response · Authors · 2022-08-05
**Look forward to further feedback**

Dear reviewers,

We thank you again for the valuable and constructive comments. We are looking forward to hearing from you about any further feedback.

If you find our response satisfactory, we hope you might view this as a sufficient reason to further raise your rating.

If you still have questions about our paper, we are willing to answer them and improve our paper.

Best, Authors

---

### Meta-Review · Area_Chair_gxA7 · 2022-08-24

**Recommendation:** Accept
**Confidence:** Certain

**Metareview:**

In this paper, the authors study the problem of robustness in image classifiers – in particular the problem of adversarial robustness. Previous work in the field of adversarial robustness focused on identifying minimal non-realistic perturbations in pixel-space that maliciously alter the classification performance of a model. In this work, the authors constrain adversarial perturbations to the space of object and camera pose that lead to poor visual recognition performance. Importantly, the space considered is constrained to be physically plausible. They leverage recent advances in Neural Rendering (NeRF) to generate realistic 3D models of objects, and optimize for non-canonical poses that lead to poor predictive performance. Finally, the authors propose a new benchmark for evaluating viewpoint robustness (ImageNet-V) which may be used to assess the general quality of any image recognition system.

The reviewers identified several notable strengths including (1) the first work to identify adversarial viewpoint as a method for assessing robustness, (2) reasonable methodology for search and optimization, (3) solid and complete experiments. The reviewers did find some weaknesses in the presentation of the material and questioning details of the experimental setup but those points were largely addressed in the responses by the authors.

Given that robustness is a very large problem much larger than the topic of image recognition, I find that the problem the authors have identified is quite important to the larger community. My only suggestion is that it would be nice if the authors showed some analysis demonstrating a positive correlation between ImageNet-V accuracy and performance on other robustness measurements. That said, given the importance of the research topic and novelty of the approach, this work will be accepted for publication at this conference.


**Award:**

Yes

---

### Decision · Program_Chairs · 2022-09-14

Accept